# Capturing Label Characteristics in VAEs

**Tom Joy**[1], **Sebastian M. Schmon**[*1,2], **Philip H. S. Torr**[1], **N. Siddharth**[*†1,3] **& Tom Rainforth**[†1]
[1]University of Oxford
[2]Improbable
[3]University of Edinburgh & The Alan Turing Institute
`tomjoy@robots.ox.ac.uk`

## Abstract

We present a principled approach to incorporating labels in variational autoencoders (VAEs) that captures the rich characteristic information associated with those labels. While prior work has typically conflated these by learning latent variables that directly correspond to label values, we argue this is contrary to the intended effect of supervision in VAEs—capturing rich label characteristics with the latents. For example, we may want to capture the characteristics of a face that make it look young, rather than just the age of the person. To this end, we develop the *characteristic capturing* VAE (CCVAE), a novel VAE model and concomitant variational objective which captures label characteristics explicitly in the latent space, eschewing direct correspondences between label values and latents. Through judicious structuring of mappings between such *characteristic latents* and labels, we show that the CCVAE can effectively learn meaningful representations of the characteristics of interest across a variety of supervision schemes. In particular, we show that the CCVAE allows for more effective and more general interventions to be performed, such as smooth traversals within the characteristics for a given label, diverse conditional generation, and transferring characteristics across datapoints.

## 1 Introduction

Learning the characteristic factors of perceptual observations has long been desired for effective machine intelligence (Brooks, 1991; Bengio et al., 2013; Hinton & Salakhutdinov, 2006; Tenenbaum, 1998). In particular, the ability to learn *meaningful* factors—capturing human-understandable characteristics from data—has been of interest from the perspective of human-like learning (Tenenbaum & Freeman, 2000; Lake et al., 2015) and improving decision making and generalization across tasks (Bengio et al., 2013; Tenenbaum & Freeman, 2000).

At its heart, learning meaningful representations of data allows one to not only make predictions, but critically also to *manipulate* factors of a datapoint. For example, we might want to manipulate the age of a person in an image. Such manipulations allow for the expression of causal effects between the meaning of factors and their corresponding realizations in the data. They can be categorized into conditional generation—the ability to construct whole exemplar data instances with characteristics dictated by constraining relevant factors—and intervention—the ability to manipulate just particular factors for a given data point, and subsequently affect only the associated characteristics.

A particularly flexible framework within which to explore the learning of meaningful representations are variational autoencoders (VAEs), a class of deep generative models where representations of data are captured in the underlying latent variables. A variety of methods have been proposed for inducing meaningful factors in this framework (Kim & Mnih, 2018; Mathieu et al., 2019; Mao et al., 2019; Kingma et al., 2014; Siddharth et al., 2017; Vedantam et al., 2018), and it has been argued that the most effective generally exploit available labels to (partially) supervise the training process (Locatello et al., 2019). Such approaches aim to associate certain factors of the representation (or equivalently factors of the generative model) with the labels, such that the former encapsulate the latter—providing a mechanism for manipulation via targeted adjustments of relevant factors.

---

[*]work done while at Oxford
[†]equal contribution

Prior approaches have looked to achieve this by directly associating certain latent variables with labels (Kingma et al., 2014; Siddharth et al., 2017; Maaløe et al., 2016). Originally motivated by the desiderata of semi–supervised classification, each label is given a corresponding latent variable of the same type (e.g. categorical), whose value is fixed to that of the label when the label is observed and imputed by the encoder when it is not.

Though natural, we argue that this assumption is not just unnecessary but actively harmful from a representation-learning perspective, particularly in the context of performing manipulations. To allow manipulations, we want to learn latent factors that capture the characteristic information *associated* with a label, which is typically much richer than just the label value itself. For example, there are various visual characteristics of people's faces associated with the label "young," but simply knowing the label is insufficient to reconstruct these characteristics for any particular instance. Learning a meaningful representation that captures these characteristics, and *isolates* them from others, requires encoding more than just the label value itself, as illustrated in Figure 1.

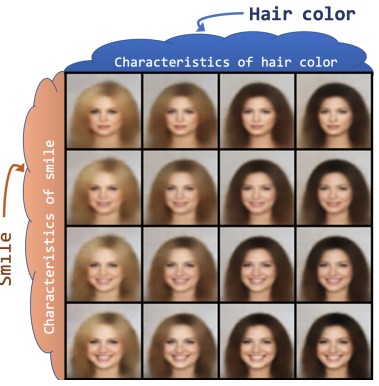

The key idea of our work is to use labels to help capture and isolate this related characteristic information in a VAE's representation. We do this by exploiting the interplay between the labels and inputs to capture more information than the labels alone convey; information that will be lost (or at least entangled) if we directly encode the label itself. Specifically, we introduce the *characteristic capturing* VAE (CCVAE) framework, which employs a novel VAE formulation which captures label characteristics explicitly in the latent space. For each label, we introduce a set of *characteristic latents* that are induced into

Figure 1: Manipulating label characteristics for "hair color" and "smile".

capturing the characteristic information associated with that label. By coupling this with a principled variational objective and carefully structuring the characteristic-latent and label variables , we show that CCVAEs successfully capture meaningful representations, enabling better performance on manipulation tasks, while matching previous approaches for prediction tasks. In particular, they permit certain manipulation tasks that cannot be performed with conventional approaches, such as manipulating characteristics *without* changing the labels themselves and producing *multiple* distinct samples consistent with the desired intervention. We summarize our contributions as follows:

 i) showing how labels can be used to capture and isolate rich *characteristic* information;
 ii) formulating CCVAEs, a novel model class and objective for supervised and semi-supervised learning in VAEs that allows this information to be captured effectively;
 iii) demonstrating CCVAEs' ability to successfully learn meaningful representations in practice.

## 2 BACKGROUND

VAEs (Kingma & Welling, 2013; Rezende et al., 2014) are a powerful and flexible class of model that combine the unsupervised representation-learning capabilities of deep autoencoders (Hinton & Zemel, 1994) with generative latent-variable models—a popular tool to capture factored low-dimensional representations of higher-dimensional observations. In contrast to deep autoencoders, generative models capture representations of data not as distinct values corresponding to observations, but rather as *distributions* of values. A generative model defines a joint distribution over observed data $x$ and latent variables $z$ as $p_\theta(x, z) = p(z)p_\theta(x \mid z)$. Given a model, learning representations of data can be viewed as performing *inference*—learning the *posterior* distribution $p_\theta(z \mid x)$ that constructs the distribution of latent values for a given observation.

VAEs employ amortized variational inference (VI) (Wainwright & Jordan, 2008; Kingma & Welling, 2013) using the encoder and decoder of an autoencoder to transform this setup by i) taking the model likelihood $p_\theta(x \mid z)$ to be parameterized by a neural network using the *decoder*, and ii) constructing an amortized variational approximation $q_\phi(z \mid x)$ to the (intractable) posterior $p_\theta(z \mid x)$ using the *encoder*. The variational approximation of the posterior enables effective estimation of the objective—maximizing the marginal likelihood—through importance sampling. The objective is obtained through invoking Jensen's inequality to derive the evidence lower bound (ELBO) of the

model which is given as:

$$\log p_\theta(\boldsymbol{x}) = \log \mathbb{E}_{q_\phi(\boldsymbol{z}|\boldsymbol{x})} \left[ \frac{p_\theta(\boldsymbol{z}, \boldsymbol{x})}{q_\phi(\boldsymbol{z} \mid \boldsymbol{x})} \right] \geq \mathbb{E}_{q_\phi(\boldsymbol{z}|\boldsymbol{x})} \left[ \log \frac{p_\theta(\boldsymbol{z}, \boldsymbol{x})}{q_\phi(\boldsymbol{z} \mid \boldsymbol{x})} \right] \equiv \mathcal{L}(\boldsymbol{x}; \phi, \theta). \quad (1)$$

Given observations $\mathcal{D} = \{\boldsymbol{x}_1, \ldots, \boldsymbol{x}_N\}$ taken to be realizations of random variables generated from an unknown distribution $p_\mathcal{D}(\boldsymbol{x})$, the overall objective is $\frac{1}{N} \sum_n \mathcal{L}(\boldsymbol{x}_n; \theta, \phi)$. Hierarchical VAEs Sønderby et al. (2016) impose a hierarchy of latent variables improving the flexibility of the approximate posterior, however we do not consider these models in this work.

Semi-supervised VAEs (SSVAEs) (Kingma et al., 2014; Maaløe et al., 2016; Siddharth et al., 2017) consider the setting where a subset of data $\mathcal{S} \subset \mathcal{D}$ is assumed to also have corresponding *labels* $\boldsymbol{y}$. Denoting the (unlabeled) data as $\mathcal{U} = \mathcal{D} \backslash \mathcal{S}$, the log-marginal likelihood is decomposed as

$$\log p(\mathcal{D}) = \sum\nolimits_{(\boldsymbol{x}, \boldsymbol{y}) \in \mathcal{S}} \log p_\theta(\boldsymbol{x}, \boldsymbol{y}) + \sum\nolimits_{\boldsymbol{x} \in \mathcal{U}} \log p_\theta(\boldsymbol{x}),$$

where the individual log-likelihoods are lower bounded by their ELBOs. Standard practice is then to treat $\boldsymbol{y}$ as a latent variable to marginalize over whenever the label is not provided. More specifically, most approaches consider splitting the latent space in $\boldsymbol{z} = \{\boldsymbol{z_y}, \boldsymbol{z}_{\backslash \boldsymbol{y}}\}$ and then directly fix $\boldsymbol{z_y} = \boldsymbol{y}$ whenever the label is provided, such that each dimension of $\boldsymbol{z_y}$ explicitly represents a predicted value of a label, with this value known exactly only for the labeled datapoints. Much of the original motivation for this (Kingma et al., 2014) was based around performing semi–supervised classification of the labels, with the encoder being used to impute the values of $\boldsymbol{z_y}$ for the unlabeled datapoints. However, the framework is also regularly used as a basis for learning meaningful representations and performing manipulations, exploiting the presence of the decoder to generate new datapoints after intervening on the labels via changes to $\boldsymbol{z_y}$. Our focus lies on the latter, for which we show this standard formulation leads to serious pathologies. Our primary goal is not to improve the fidelity of generations, but instead to demonstrate how label information can be used to structure the latent space such that it encapsulates and disentangles the characteristics associated with the labels.

## 3 RETHINKING SUPERVISION

As we explained in the last section, the de facto assumption for most approaches to supervision in VAEs is that the labels correspond to a partially observed augmentation of the latent space, $\boldsymbol{z_y}$. However, this can cause a number of issues if we want the latent space to encapsulate not just the labels themselves, but also the characteristics *associated* with these labels. For example, encapsulating the youthful characteristics of a face, not just the fact that it is a "young" face. At an abstract level, such an approach fails to capture the relationship between the inputs and labels: it fails to isolate characteristic information associated with each label from the other information required to reconstruct data. More specifically, it fails to deal with the following issues.

Firstly, the information in a datapoint associated with a label is richer than stored by the (typically categorical) label itself. That is not to say such information is absent when we impose $\boldsymbol{z_y} = \boldsymbol{y}$, but here it is *entangled* with the other latent variables $\boldsymbol{z}_{\backslash \boldsymbol{y}}$, which simultaneously contain the associated information for *all* the labels. Moreover, when $\boldsymbol{y}$ is categorical, it can be difficult to ensure that the VAE actually uses $\boldsymbol{z_y}$, rather than just capturing information relevant to reconstruction in the higher-capacity, continuous, $\boldsymbol{z}_{\backslash \boldsymbol{y}}$. Overcoming this is challenging and generally requires additional heuristics and hyper-parameters.

Second, we may wish to manipulate characteristics without fully changing the categorical label itself. For example, making a CelebA image depict more or less 'smiling' without fully changing its "smile" label. Here we do not know how to manipulate the latents to achieve this desired effect: we can only do the binary operation of changing the relevant variable in $\boldsymbol{z_y}$. Also, we often wish to keep a level of diversity when carrying out conditional generation and, in particular, interventions. For example, if we want to add a smile, there is no single correct answer for how the smile would look, but taking $\boldsymbol{z_y} = $ "smile" only allows for a single point estimate for the change.

Finally, taking the labels to be explicit latent variables can cause a mismatch between the VAE prior $p(\boldsymbol{z})$ and the pushforward distribution of the data to the latent space $q(\boldsymbol{z}) = \mathbb{E}_{p_\mathcal{D}(\boldsymbol{x})}[q_\phi(\boldsymbol{z} \mid \boldsymbol{x})]$. During training, latents are effectively generated according to $q(\boldsymbol{z})$, but once learned, $p(\boldsymbol{z})$ is used to make generations; variations between the two effectively corresponds to a train-test mismatch. As there is a ground truth data distribution over the labels (which are typically not independent), taking the latents as the labels themselves implies that there will be a ground truth $q(\boldsymbol{z_y})$. However, as this is not generally known a priori, we will inevitably end up with a mismatch.

**What do we want from supervision?**    Given these issues, it is natural to ask whether having latents directly correspond to labels is actually necessary. To answer this, we need to think about exactly what it is we are hoping to achieve through the supervision itself. Along with uses of VAEs more generally, the three most prevalent tasks are: **a) Classification**, predicting the labels of inputs where these are not known a priori; **b) Conditional Generation**, generating new examples conditioned on those examples conforming to certain desired labels; and **c) Intervention**, manipulating certain desired characteristics of a data point before reconstructing it.

Inspecting these tasks, we see that for classification we need a classifier form $z$ to $y$, for conditional generation we need a mechanism for sampling $z$ given $y$, and for inventions we need to know how to manipulate $z$ to bring about a desired change. None of these require us to have the labels directly correspond to latent variables. Moreover, as we previously explained, this assumption can be actively harmful, such as restricting the range of interventions that can be performed.

## 4    CHARACTERISTIC CAPTURING VARIATIONAL AUTOENCODERS

To correct the issues discussed in the last section, we suggest eschewing the treatment of labels as direct components of the latent space and instead employ them to condition latent variables which are designed to capture the characteristics. To this end, we similarly split the latent space into two components, $z = \{z_c, z_{\setminus c}\}$, but where $z_c$, the *characteristic latent*, is now designed to capture the characteristics associated with labels, rather than directly encode the labels themselves. In this breakdown, $z_{\setminus c}$ is intended only to capture information not directly associated with any of the labels, unlike $z_{\setminus y}$ which was still tasked with capturing the characteristic information.

For the purposes of exposition and purely to demonstrate how one might apply this schema, we first consider a standard VAE, with a latent space $z = \{z_c, z_{\setminus c}\}$. The latent representation of the VAE will implicitly contain characteristic information required to perform classification, however the structure of the latent space will be arranged to optimize for reconstruction and characteristic information may be *entangled* between $z_c$ and $z_{\setminus c}$. If we were now to jointly learn a classifier—from $z_c$ to $y$—with the VAE, resulting in the following objective:

$$\mathcal{J} = \sum_{\boldsymbol{x} \in \mathcal{U}} \mathcal{L}_{\text{VAE}}(\boldsymbol{x}) + \sum_{(\boldsymbol{x}, \boldsymbol{y}) \in \mathcal{S}} \left( \mathcal{L}_{\text{VAE}}(\boldsymbol{x}) + \alpha \mathbb{E}_{q_\phi(\boldsymbol{z}|\boldsymbol{x})} \left[ \log q_\varphi(\boldsymbol{y} \mid \boldsymbol{z}_c) \right] \right), \qquad (2)$$

where $\alpha$ is a hyperparameter, there will be pressure on the encoder to place characteristic information in $z_c$, which can be interpreted as a stochastic layer containing the information needed for classification *and* reconstruction[1]. The classifier thus acts as a tool allowing $y$ to influence the structure of $z$, it is this high level concept, i.e. using $y$ to structure $z$, that we utilize in this work.

However, in general, the characteristics of different labels will be *entangled* within $z_c$. Though it will contain the required information, the latents will typically be uninterpretable, and it is unclear how we could perform conditional generation or interventions. To *disentangle* the characteristics of different labels, we further partition the latent space, such that the classification of particular labels $y^i$ only has access to particular latents $z_c^i$ and thus $\log q_\varphi(\boldsymbol{y} \mid \boldsymbol{z}_c) = \sum_i \log q_{\varphi^i}(y^i \mid \boldsymbol{z}_c^i)$. This has the critical effect of forcing the characteristic information needed to classify $y^i$ to be stored only in the corresponding $z_c^i$, providing a means to encapsulate such information for each label separately. We further see that it addresses many of the prior issues: there are no measure-theoretic issues as $z_c^i$ is not discrete, diversity in interventions is achieved by sampling different $z_c^i$ for a given label, $z_c^i$ can be manipulated while remaining within class decision boundaries, and a mismatch between $p(z_c)$ and $q(z_c)$ does not manifest as there is no ground truth for $q(z_c)$.

How to conditionally generate or intervene when training with (2) is not immediately obvious though. However, the classifier *implicitly* contains the requisite information to do this via *inference* in an implied Bayesian model. For example, conditional generation needs samples from $p(z_c)$ that classify to the desired labels, e.g. through rejection sampling. See Appendix A for further details.

### 4.1    THE CHARACTERISTIC CAPTURING VAE

One way to address the need for inference is to introduce a conditional generative model $p_\psi(\boldsymbol{z}_c \mid \boldsymbol{y})$, simultaneously learned alongside the classifier introduced in (2), along with a prior $p(\boldsymbol{y})$. This

---

[1]Though, for convenience, we implicitly assume here, and through the rest of the paper, that the labels are categorical such that the mapping $z_c \to y$ is a classifier, we note that the ideas apply equally well if some labels are actually continuous, such that this mapping is now a probabilistic regression.

approach, which we term the CCVAE, allows the required sampling for conditional generations and interventions directly. Further, by persisting with the latent partitioning above, we can introduce a factorized set of generative models $p(\boldsymbol{z}_c \mid \boldsymbol{y}) = \prod_i p(\boldsymbol{z}_c^i \mid y^i)$, enabling easy generation and manipulation of $\boldsymbol{z}_c^i$ individually. CCVAE ensures that labels remain a part of the model for unlabeled datapoints, which transpires to be important for effective learning in practice.

To address the issue of learning, we perform variational inference, treating $\boldsymbol{y}$ as a partially observed auxiliary variable. The final graphical model is illustrated in Figure 2. The CCVAE can be seen as a way of combining top-down and bottom-up information to obtain a structured latent representation. However, it is important to highlight that CCVAE does not contain a hierarchy of latent variables. Unlike a hierarchical VAE, reconstruction is performed only from $\boldsymbol{z} \sim q_\phi(\boldsymbol{z} \mid \boldsymbol{x})$ *without* going through the "deeper" $\boldsymbol{y}$, as doing so would lead to a loss of information due to the bottleneck of $\boldsymbol{y}$. By enforcing each label variable to link to different characteristic-latent dimensions, we are able to isolate the generative factors corresponding to different label characteristics.

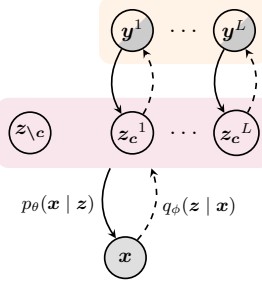

Figure 2: CCVAE graphical model.

### 4.2 MODEL OBJECTIVE

We now construct an objective function that encapsulates the model described above, by deriving a lower bound on the full model log-likelihood which factors over the supervised and unsupervised subsets as discussed in § 2. The supervised objective can be defined as

$$\log p_{\theta,\psi}(\boldsymbol{x}, \boldsymbol{y}) \geq \mathbb{E}_{q_{\varphi,\phi}(\boldsymbol{z}|\boldsymbol{x},\boldsymbol{y})} \left[ \log \frac{p_\theta(\boldsymbol{x} \mid \boldsymbol{z}) p_\psi(\boldsymbol{z} \mid \boldsymbol{y}) p(\boldsymbol{y})}{q_{\varphi,\phi}(\boldsymbol{z} \mid \boldsymbol{x}, \boldsymbol{y})} \right] \equiv \mathcal{L}_{\text{CCVAE}}(\boldsymbol{x}, \boldsymbol{y}), \qquad (3)$$

with $p_\psi(\boldsymbol{z} \mid \boldsymbol{y}) = p(\boldsymbol{z}_{\backslash c}) p_\psi(\boldsymbol{z}_{\boldsymbol{c}} \mid \boldsymbol{y})$. Here, we avoid directly modeling $q_{\varphi,\phi}(\boldsymbol{z} \mid \boldsymbol{x}, \boldsymbol{y})$; instead leveraging the conditional independence $\boldsymbol{x} \perp\!\!\!\perp \boldsymbol{y} \mid \boldsymbol{z}$, along with Bayes rule, to give

$$q_{\varphi,\phi}(\boldsymbol{z} \mid \boldsymbol{x}, \boldsymbol{y}) = \frac{q_\varphi(\boldsymbol{y} \mid \boldsymbol{z}_c) q_\phi(\boldsymbol{z} \mid \boldsymbol{x})}{q_{\varphi,\phi}(\boldsymbol{y} \mid \boldsymbol{x})}, \quad \text{where} \quad q_{\varphi,\phi}(\boldsymbol{y} \mid \boldsymbol{x}) = \int q_\varphi(\boldsymbol{y} \mid \boldsymbol{z}_c) q_\phi(\boldsymbol{z} \mid \boldsymbol{x}) \mathrm{d}\boldsymbol{z}.$$

Using this equivalence in (3) yields (see Appendix B.1 for a derivation and numerical details)

$$\mathcal{L}_{\text{CCVAE}}(\boldsymbol{x}, \boldsymbol{y}) = \mathbb{E}_{q_\phi(\boldsymbol{z}|\boldsymbol{x})} \left[ \frac{q_\varphi(\boldsymbol{y} \mid \boldsymbol{z}_c)}{q_{\varphi,\phi}(\boldsymbol{y} \mid \boldsymbol{x})} \log \frac{p_\theta(\boldsymbol{x} \mid \boldsymbol{z}) p_\psi(\boldsymbol{z} \mid \boldsymbol{y})}{q_\varphi(\boldsymbol{y} \mid \boldsymbol{z}_c) q_\phi(\boldsymbol{z} \mid \boldsymbol{x})} \right] + \log q_{\varphi,\phi}(\boldsymbol{y} \mid \boldsymbol{x}) + \log p(\boldsymbol{y}). \quad (4)$$

Note that a classifier term $\log q_{\varphi,\phi}(\boldsymbol{y} \mid \boldsymbol{x})$ falls out naturally from the derivation, unlike previous models (e.g. Kingma et al. (2014); Siddharth et al. (2017)). Not placing the labels directly in the latent space is crucial for this feature. When defining latents to directly correspond to labels, observing both $\boldsymbol{x}$ and $\boldsymbol{y}$ *detaches* the mapping $q_{\varphi,\phi}(\boldsymbol{y} \mid \boldsymbol{x})$ between them, resulting in the parameters $(\varphi, \phi)$ not being learned—motivating addition of an explicit (weighted) classifier. Here, however, observing both $\boldsymbol{x}$ and $\boldsymbol{y}$ does not detach any mapping, since they are always connected via an unobserved random variable $\boldsymbol{z}_c$, and hence do not need additional terms. From an implementation perspective, this classifier strength can be increased, we experimented with this, but found that adjusting the strength had little effect on the overall classification accuracies. We consider this insensitivity to be a significant strength of this approach, as the model is able to apply enough pressure to the latent space to obtain high classification accuracies without having to hand tune parameter values. We find that the gradient norm of the classifier parameters suffers from a high variance during training, we find that not reparameterizing through $\boldsymbol{z}_c$ in $q_\varphi(\boldsymbol{y} \mid \boldsymbol{z}_c)$ reduces this affect and aides training, see Appendix C.3.1 for details.

For the datapoints without labels, we can again perform variational inference, treating the labels as random variables. Specifically, the unsupervised objective, $\mathcal{L}_{\text{CCVAE}}(\boldsymbol{x})$, derives as the standard (unsupervised) ELBO. However, it requires marginalising over labels as $p(\boldsymbol{z}) = p(\boldsymbol{z}_c) p(\boldsymbol{z}_{\backslash c}) = p(\boldsymbol{z}_{\backslash c}) \sum_{\boldsymbol{y}} p(\boldsymbol{z}_c|\boldsymbol{y}) p(\boldsymbol{y})$. This can be computed exactly, but doing so can be prohibitively expensive if the number of possible label combinations is large. In such cases, we apply Jensen's inequality a second time to the expectation over $\boldsymbol{y}$ (see Appendix B.2) to produce a looser, but cheaper to calculate, ELBO given as

$$\mathcal{L}_{\text{CCVAE}}(\boldsymbol{x}) = E_{q_\phi(\boldsymbol{z}|\boldsymbol{x}) q_\varphi(\boldsymbol{y}|\boldsymbol{z}_c)} \left[ \log \left( \frac{p_\theta(\boldsymbol{x} \mid \boldsymbol{z}) p_\psi(\boldsymbol{z} \mid \boldsymbol{y}) p(\boldsymbol{y})}{q_\varphi(\boldsymbol{y} \mid \boldsymbol{z}_c) q_\phi(\boldsymbol{z} \mid \boldsymbol{x})} \right) \right]. \qquad (5)$$

Combining (4) and (5), we get the following lower bound on the log probability of the data

$$\log p(\mathcal{D}) \geq \sum_{(\boldsymbol{x}, \boldsymbol{y}) \in \mathcal{S}} \mathcal{L}_{\text{CCVAE}}(\boldsymbol{x}, \boldsymbol{y}) + \sum_{\boldsymbol{x} \in \mathcal{U}} \mathcal{L}_{\text{CCVAE}}(\boldsymbol{x}), \qquad (6)$$

that unlike prior approaches faithfully captures the variational free energy of the model. As shown in § 6, this enables a range of new capabilities and behaviors to encapsulate label characteristics.

## 5 RELATED WORK

The seminal work of Kingma et al. (2014) was the first to consider supervision in the VAEs setting, introducing the M2 model for semi–supervised classification which was also approach to place labels directly in the latent space. The related approach of Maaløe et al. (2016) augments the encoding distribution with an additional, unobserved latent variable, enabling better semi-supervised classification accuracies. Siddharth et al. (2017) extended the above work to automatically derive the regularised objective for models with arbitrary (pre-defined) latent dependency structures. The approach of placing labels directly in the latent space was also adopted in Li et al. (2019). Regarding the disparity between continuous and discrete latent variables in the typical semi-supervised VAEs, Dupont (2018) provide an approach to enable effective *unsupervised* learning in this setting.

From a purely modeling perspective, there also exists prior work on VAEs involving hierarchies of latent variables, exploring richer higher-order inference and issues with redundancy among latent variables both in unsupervised (Ranganath et al., 2016; Zhao et al., 2017) and semi-supervised (Maaløe et al., 2017; 2019) settings. In the unsupervised case, these hierarchical variables do not have a direct interpretation, but exist merely to improve the flexibility of the encoder. The semi-supervised approaches extend the basic M2 model to hierarchical VAEs by incorporating the labels as an additional latent (see Appendix F in Maaløe et al., 2019, for example), and hence must incorporate additional regularisers in the form of classifiers as in the case of M2. Moreover, by virtue of the typical dependencies assumed between labels and latents, it is difficult to disentangle the characteristics just associated with the label from the characteristics associated with the rest of the data—something we capture using our simpler split latents $(\boldsymbol{z_c}, \boldsymbol{z_{\backslash c}})$.

From a more conceptual standpoint, Mueller et al. (2017) introduces interventions (called revisions) on VAEs for text data, regressing to auxiliary sentiment scores as a means of influencing the latent variables. This formulation is similar to (2) in spirit, although in practice they employ a range of additional factoring and regularizations particular to their domain of interest, in addition to training models in stages, involving different objective terms. Nonetheless, they share our desire to enforce meaningfulness in the latent representations through auxiliary supervision.

Another related approach involves explicitly treating labels as another data *modality* (Vedantam et al., 2018; Suzuki et al., 2017; Wu & Goodman, 2018; Shi et al., 2019). This work is motivated by the need to learn latent representations that *jointly encode* data from different modalities. Looking back to (3), by refactoring $p(\boldsymbol{z} \mid \boldsymbol{y})p(\boldsymbol{y})$ as $p(\boldsymbol{y} \mid \boldsymbol{z})p(\boldsymbol{z})$, and taking $q(\boldsymbol{z} \mid \boldsymbol{x}, \boldsymbol{y}) = \mathcal{G}(q(\boldsymbol{z} \mid \boldsymbol{x}), q(\boldsymbol{z} \mid \boldsymbol{y}))$, one derives *multi-modal* VAEs, where $\mathcal{G}$ can construct a product (Wu & Goodman, 2018) or mixture (Shi et al., 2019) of experts. Of these, the MVAE (Wu & Goodman, 2018) is more closely related to our setup here, as it explicitly targets cases where alternate data modalities are labels. However, they differ in that the latent representations are not structured explicitly to map to distinct classifiers, and do not explore the question of explicitly capturing the label characteristics. The JLVM model of Adel et al. (2018) is similar to the MVAE, but is motivated from an interpretability perspective—with labels providing 'side-channel' information to constrain latents. They adopt a flexible normalising-flow posterior from data $\boldsymbol{x}$, along with a multi-component objective that is additionally regularised with the information bottleneck between data $\boldsymbol{x}$, latent $\boldsymbol{z}$, and label $\boldsymbol{y}$.

DIVA (Ilse et al., 2019) introduces a similar graphical model to ours, but is motivated to learn a generalized classifier for different domains. The objective is formed of a classifier which is regularized by a variational term, requiring additional hyper-parameters and preventing the ability to disentangle the representations. In Appendix C.4 we propose some modifications to DIVA that allow it to be applied in our problem domain.

In terms of interoperability, the work of Ainsworth et al. (2018) is closely related to ours, but they focus primarily on group data and not introducing labels. Here the authors employ sparsity in the multiple linear transforms for each decoder (one for each group) to encourage certain latent dimensions to encapsulate certain factors in the sample, thus introducing interoperability into the

model. Tangentially to VAEs, similar objectives of structuring the latent space using GANs also exist Xiao et al. (2017; 2018), although they focus purely on interventions and cannot perform conditional generations, classification, or estimate likelihoods.

## 6 EXPERIMENTS

Following our reasoning in § 3 we now showcase the efficacy of CCVAE for the three broad aims of *(a) intervention*, *(b) conditional generation* and *(c) classification* for a variety of supervision rates, denoted by $f$. Specifically, we demonstrate that CCVAE is able to: encapsulate characteristics for each label in an isolated manner; introduce diversity in the conditional generations; permit a finer control on interventions; and match traditional metrics of baseline models. Furthermore, we demonstrate that no existing method is able to perform all of the above,[2] highlighting its sophistication over existing methods. We compare against: M2 (Kingma et al., 2014); MVAE (Wu & Goodman, 2018); and our modified version of DIVA (Ilse et al., 2019). See Appendix C.4 for details.

To demonstrate the capture of label characteristics, we consider the multi-label setting and utilise the Chexpert (Irvin et al., 2019) and CelebA (Liu et al., 2015) datasets.[3] For CelebA, we restrict ourselves to the 18 labels which are distinguishable in reconstructions; see Appendix C.1 for details. We use the architectures from Higgins et al. (2016) for the encoder and decoder. The label-predictive distribution $q_\varphi(\boldsymbol{y} \mid \boldsymbol{z}_c)$ is defined as $\mathrm{Ber}(\boldsymbol{y} \mid \boldsymbol{\pi}_\varphi(\boldsymbol{z}_c))$ with a diagonal transformation $\boldsymbol{\pi}_\varphi(\cdot)$ enforcing $q_\varphi(\boldsymbol{y} \mid \boldsymbol{z}_c) = \prod_i q_{\varphi^i}(y_i \mid \boldsymbol{z}_c{}^i)$. The conditional prior $p_\psi(\boldsymbol{z}_c \mid \boldsymbol{y})$ is then defined as $\mathcal{N}(\boldsymbol{z}_c | \boldsymbol{\mu}_\psi(\boldsymbol{y}), \mathrm{diag}(\boldsymbol{\sigma}_\psi^2(\boldsymbol{y})))$ with appropriate factorization, and has its parameters also derived through MLPs. See Appendix C.3 for further details.

### 6.1 INTERVENTIONS

If CCVAE encapsulates characteristics of a label in a single latent (or small set of latents), then it should be able to smoothly manipulate these characteristics without severely affecting others. This allows for finer control during interventions, which is not possible when the latent variables directly correspond to labels. To demonstrate this, we traverse two dimensions of the latent space and display the reconstructions in Figure 3. These examples indicate that CCVAE is indeed able to smoothly manipulate characteristics. For example, in **b)** we are able to induce varying skin tones rather than have this be a binary intervention on `pale skin`, unlike DIVA in **a)**. In **c)**, the $\boldsymbol{z}_c^i$ associated with the `necktie` label has also managed to encapsulate information about whether someone is wearing a shirt or is bare-necked. No such traversals are possible for M2 and it is not clear how one would do them for MVAE; additional results, including traversals for DIVA, are given in Appendix D.2.

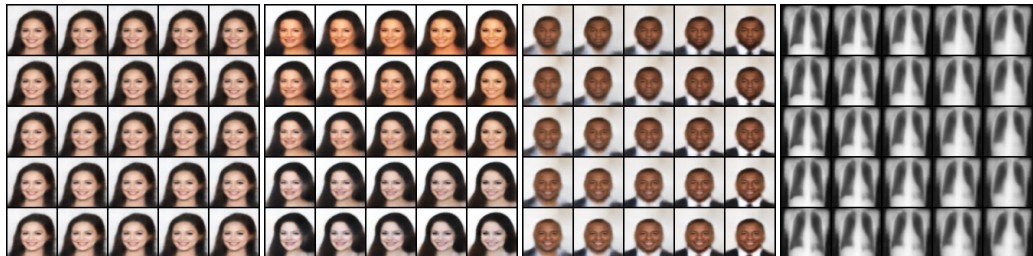

Figure 3: Continuous interventions through traversal of $\boldsymbol{z}_c$. From left to right, a) DIVA `pale skin` and `young`; b) CCVAE `pale skin` and `young`; c) CCVAE `smiling` and `necktie`; d) CCVAE `Pleural Effusion` and `Cardiomegaly`.

### 6.2 DIVERSITY OF GENERATIONS

Label characteristics naturally encapsulate diversity (e.g. there are many ways to smile) which should be present in the learned representations. By virtue of the structured mappings between labels and characteristic latents, and since $\boldsymbol{z}_c$ is parameterized by continuous distributions, CCVAE is able to capture diversity in representations, allowing exploration for an attribute (e.g. smile) while

---

[2]DIVA can perform the same tasks as CCVAE but only with the modifications we ourselves suggest and still not to a comparable quality.

[3]CCVAE is well-suited to multi-label problems, but also works on multi-class problems. See Appendix D.6 for results and analyses on MNIST and FashionMNIST.

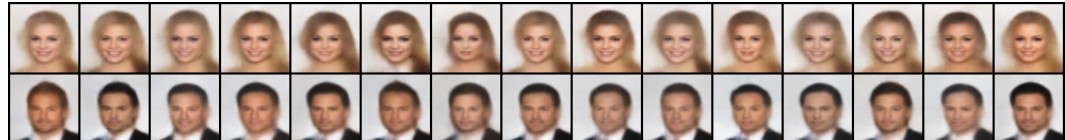

Figure 4: Diverse conditional generations for CCVAE, $\boldsymbol{y}$ is held constant along each row and each column represents a different sample for $\boldsymbol{z_c} \sim p(\boldsymbol{z_c}|\boldsymbol{y})$. $\boldsymbol{z_{\backslash c}}$ is held constant over the entire figure.

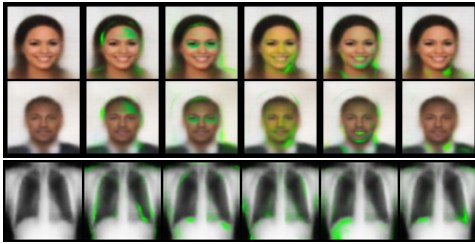

Figure 5: Variance in reconstructions when intervening on a single label. [Top two] CelebA, from left to right: reconstruction, `bangs, eyeglasses, pale skin, smiling, necktie.`. [Bottom] Chexpert: reconstruction, `cardiomegaly, edema, consolidation, atelectasis, pleural effusion`.

preserving other characteristics. This is not possible with labels directly defined as latents, as only discrete choices can be made—diversity can only be introduced here by sampling from the unlabeled latent space—which necessarily affects all other characteristics. To demonstrate this, we reconstruct multiple times with $\boldsymbol{z} = \{\boldsymbol{z_c} \sim p_\psi(\boldsymbol{z_c} \mid \boldsymbol{y}), \boldsymbol{z_{\backslash c}}\}$ for a fixed $\boldsymbol{z_{\backslash c}}$. We provide qualitative results in Figure 4.

If several samples are taken from $\boldsymbol{z_c} \sim p_\psi(\boldsymbol{z_c} \mid \boldsymbol{y})$ when intervening on only a single characteristic, the resulting variations in pixel values should be focused around the locations relevant to that characteristic, e.g. pixel variations should be focused around the neck when intervening on `necktie`. To demonstrate this, we perform single interventions on each class, and take multiple samples of $\boldsymbol{z_c} \sim p_\psi(\boldsymbol{z_c} \mid \boldsymbol{y})$. We then display the variance of each pixel in the reconstruction in green in Figure 5, where it can be seen that generally there is only variance in the spatial locations expected. Interestingly, for the class `smile` (2nd from right), there is variance in the jaw line, suggesting that the model is able capture more subtle components of variation that just the mouth.

### 6.3 CLASSIFICATION

To demonstrate that reparameterizing the labels in the latent space does not hinder classification accuracy, we inspect the predictive ability of CCVAE across a range of supervision rates, given in Table 1. It can be observed that CCVAE generally obtains prediction accuracies slightly superior to other models. We emphasize here that CCVAE's primary purpose is not to achieve better classification accuracies; we are simply checking that it does not harm them, which it most clearly does not.

Table 1: Classification accuracies.

| | CelebA | | | | Chexpert | | | |
|---|---|---|---|---|---|---|---|---|
| Model | $f = 0.004$ | $f = 0.06$ | $f = 0.2$ | $f = 1.0$ | $f = 0.004$ | $f = 0.06$ | $f = 0.2$ | $f = 1.0$ |
| CCVAE | **0.832** | **0.862** | **0.878** | **0.900** | **0.809** | **0.792** | **0.794** | **0.826** |
| M2 | 0.794 | 0.862 | 0.877 | 0.893 | 0.799 | 0.779 | 0.777 | 0.774 |
| DIVA | 0.807 | 0.860 | 0.867 | 0.877 | 0.747 | 0.786 | 0.781 | 0.775 |
| MVAE | 0.793 | 0.828 | 0.847 | 0.864 | 0.759 | 0.787 | 0.767 | 0.715 |

### 6.4 DISENTANGLEMENT OF LABELED AND UNLABELED LATENTS

If a model can correctly disentangle the label characteristics from other generative factors, then manipulating $\boldsymbol{z_{\backslash c}}$ should not change the label characteristics of the reconstruction. To demonstrate this, we perform "characteristic swaps," where we first obtain $\boldsymbol{z} = \{\boldsymbol{z_c}, \boldsymbol{z_{\backslash c}}\}$ for a given image, then swap in the characteristics $\boldsymbol{z_c}$ to another image before reconstructing. This should apply the exact characteristics, not just the label, to the scene/background of the other image (cf. Figure 6).

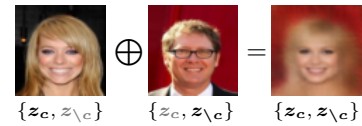

$\{\pmb{z_c}, z_{\backslash c}\}$  $\{z_c, \pmb{z_{\backslash c}}\}$  $\{\pmb{z_c}, \pmb{z_{\backslash c}}\}$

Figure 6: Characteristic swap, where the characteristics of the first image (`blond hair`, `smiling`, `heavy makeup`, `female`, `no necktie`, `no glasses` etc.) are transfered to the unlabeled characteristics of the second (`red background` etc.).

Comparing CCVAE to our baselines in Figure 7, we see that CCVAE is able to transfer the exact characteristics to a greater extent than other models. Particular attention is drawn to the preservation of labeled characteristics in each row, where CCVAE is able to preserve characteristics, like the precise skin tone and hair color of the pictures on the left. We see that M2 is only able to preserve the label and not the exact characteristic, while MVAE performs very poorly, effectively ignoring the attributes entirely. Our modified DIVA variant performs reasonably well, but less reliably and at the cost of reconstruction fidelity compared to CCVAE.

unlabeled contextual attributes, $z_{\backslash c}$

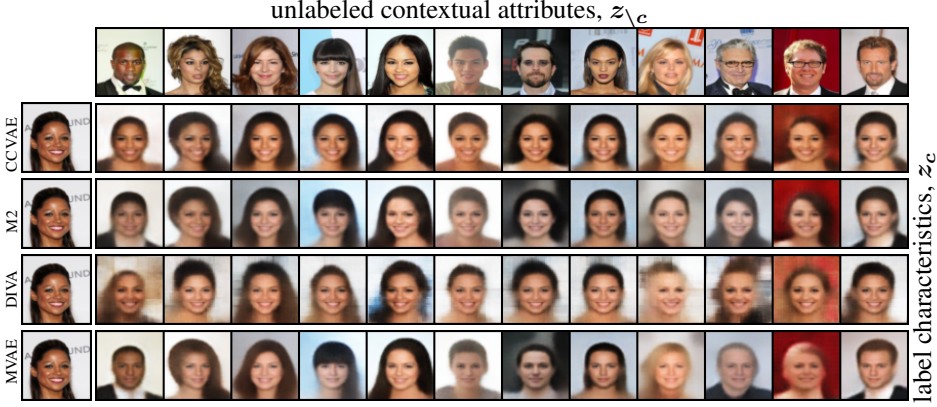

label characteristics, $z_c$

Figure 7: Characteristic swaps. Characteristics (`smiling`, `brown hair`, `skin tone`, etc) of the left image should be preserved along the row while background information should be preserved along the column.

An ideal characteristic swap should not change the probability assigned by a pre-trained classifier between the original image and a swapped one. We employ this as a quantitative measure, reporting the average difference in log probabilities for multiple swaps in Table 2. CCVAE is able to preserve the characteristics to a greater extent than other models. DIVA's performance is largely due to its heavier weighting on the classifier, which adversely affects reconstructions, as seen earlier.

Table 2: Difference in log-probabilities of pre-trained classifier from denotation swaps, lower is better.

| | CelebA | | | | Chexpert | | | |
|---|---|---|---|---|---|---|---|---|
| Model | $f = 0.004$ | $f = 0.06$ | $f = 0.2$ | $f = 1.0$ | $f = 0.004$ | $f = 0.06$ | $f = 0.2$ | $f = 1.0$ |
| CCVAE | **1.177** | **0.890** | **0.790** | **0.758** | **1.142** | **1.221** | **1.078** | **1.084** |
| M2 | 2.118 | 1.194 | 1.179 | 1.143 | 1.624 | 1.43 | 1.41 | 1.415 |
| DIVA | 1.489 | 0.976 | 0.996 | 0.941 | 1.36 | 1.25 | 1.199 | 1.259 |
| MVAE | 2.114 | 2.113 | 2.088 | 2.121 | 1.618 | 1.624 | 1.618 | 1.601 |

## 7  DISCUSSION

We have presented a novel mechanism for faithfully capturing label characteristics in VAEs, the *characteristic capturing* VAE (CCVAE), which captures label characteristics explicitly in the latent space while eschewing direct correspondences between label values and latents. This has allowed us to encapsulate and disentangle the *characteristics* associated with labels, rather than just the label values. We are able to do so without affecting the ability to perform the tasks one typically does in the (semi-)supervised setting—namely classification, conditional generation, and intervention. In particular, we have shown that, not only does this lead to more effective conventional label-switch interventions, it also allows for more fine-grained interventions to be performed, such as producing diverse sets of samples consistent with an intervened label value, or performing characteristic swaps between datapoints that retain relevant features.

## 8 ACKNOWLEDGMENTS

TJ, PHST, and NS were supported by the ERC grant ERC-2012-AdG 321162-HELIOS, EPSRC grant Seebibyte EP/M013774/1 and EPSRC/MURI grant EP/N019474/1. Toshiba Research Europe also support TJ. TJ would also like to thank Dr. M. Stoddart. PHST would also like to acknowledge the Royal Academy of Engineering and FiveAI.

SMS was partially supported by the Engineering and Physical Sciences Research Council (EPSRC) grant EP/K503113/1.

TR's research leading to these results has received funding from a Christ Church Oxford Junior Research Fellowship and from Tencent AI Labs.

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

## A    CONDITIONAL GENERATION AND INTERVENTION FOR EQUATION (2)

For the model trained using (2) as the objective to be usable, we must consider whether it can carry out the classification, conditional generation, and intervention tasks outlined previously. Of these, classification is straightforward, but it is less apparent how the others could be performed. The key here is to realize that the classifier itself *implicitly* contains the information required to perform these tasks.

Consider first conditional generation and note that we still have access to the prior $p(\boldsymbol{z})$ as per a standard VAE. One simple way of performing conditional generation would be to conduct a rejection sampling where we draw samples $\hat{\boldsymbol{z}} \sim p(\boldsymbol{z})$ and then accept these if and only if they lead to the classifier predicting the desired labels up to a desired level of confidence, i.e. $q_\phi(\boldsymbol{y} \mid \hat{\boldsymbol{z}}_c) > \lambda$ where $0 < \lambda < 1$ is some chosen confidence threshold. Though such an approach is likely to be highly inefficient for any general $p(\boldsymbol{z})$ due to the curse of dimensionality, in the standard setting where each dimension of $\boldsymbol{z}$ is independent, this rejection sampling can be performed separately for each $\boldsymbol{z}_c^i$, making it relatively efficient. More generally, we have that conditional generation becomes an inference problem where we wish to draw samples from

$$p\left(\boldsymbol{z} \mid \{q_\phi(\boldsymbol{y} \mid \boldsymbol{z}_c) > \lambda\}\right) \propto p(\boldsymbol{z})\mathbb{I}\left(q_\phi(\boldsymbol{y} \mid \boldsymbol{z}_c) > \lambda\right).$$

Interventions can also be performed in an analogous manner. Namely, for a conventional intervention where we change one or more labels, we can simply resample the $\boldsymbol{z}_c^i$ associated we those labels, thereby sampling new characteristics to match the new labels. Further, unlike prior approaches, we can perform alternative interventions too. For example, we might attempt to find the closest $\boldsymbol{z}_c^i$ to the original that leads to the class label changing; this can be done in a manner akin to how adversarial attacks are performed. Alternatively, we might look to manipulate the $\boldsymbol{z}_c^i$ without actually changing the class itself to see what other characteristics are consistent with the labels.

To summarize, (2) yields an objective which provides a way of learning a semi-supervised VAEs that avoids the pitfalls of directly fixing the latents to correspond to labels. It still allows us to perform all the tasks usually associated with semi-supervised VAEs and in fact allows a more general form of interventions to be performed. However, this comes at the cost of requiring inference to perform conditional generation or interventions. Further, as the label variables $\boldsymbol{y}$ are absent when the labels are unobserved, there may be empirical complications with forcing all the denotational information to be encoded to the appropriate characteristic latent $\boldsymbol{z}_c^i$. In particular, we still have a hyperparameter $\alpha$ that must be carefully tuned to ensure the appropriate balance between classification and reconstruction.

## B    MODEL FORMULATION

### B.1    VARIATIONAL LOWER BOUND

In this section we provide the mathematical details of our objective functions. We show how to derive it as a lower bound to the marginal model likelihood and show how we estimate the model components.

The variational lower bound for the generative model in Figure 2, is given as

$$\mathcal{L}_{\text{CCVAE}} = \sum_{\boldsymbol{x} \in \mathcal{U}} \mathcal{L}_{\text{CCVAE}}(\boldsymbol{x}) + \sum_{(\boldsymbol{x},\boldsymbol{y}) \in \mathcal{S}} \mathcal{L}_{\text{CCVAE}}(\boldsymbol{x}, \boldsymbol{y})$$

$$\mathcal{L}_{\text{CCVAE}}(\boldsymbol{x}, \boldsymbol{y}) = E_{q_\phi(\boldsymbol{z}|\boldsymbol{x})} \left[ \frac{q_\varphi(\boldsymbol{y} \mid \boldsymbol{z}_c)}{q_{\varphi,\phi}(\boldsymbol{y} \mid \boldsymbol{x})} \log \left( \frac{p_\theta(\boldsymbol{x} \mid \boldsymbol{z})p_\psi(\boldsymbol{z} \mid \boldsymbol{y})}{q_\varphi(\boldsymbol{y} \mid \boldsymbol{z}_c)q_\phi(\boldsymbol{z} \mid \boldsymbol{x})} \right) \right] + \log q_{\varphi,\phi}(\boldsymbol{y} \mid \boldsymbol{x}) + \log p(\boldsymbol{y}),$$

$$\mathcal{L}_{\text{CCVAE}}(\boldsymbol{x}) = E_{q_\phi(\boldsymbol{z}|\boldsymbol{x})q_\varphi(\boldsymbol{y}|\boldsymbol{z}_c)} \left[ \log \left( \frac{p_\theta(\boldsymbol{x} \mid \boldsymbol{z})p_\psi(\boldsymbol{z}_c \mid \boldsymbol{y})p(\boldsymbol{y})}{q_\varphi(\boldsymbol{y} \mid \boldsymbol{z}_c)q_\phi(\boldsymbol{z} \mid \boldsymbol{x})} \right) \right].$$

The overall likelihood in the semi-supervised case is given as

$$p_\theta(\boldsymbol{x}, \boldsymbol{y}) = \prod_{(\boldsymbol{x},\boldsymbol{y}) \in \mathcal{S}} p_\theta(\boldsymbol{x}, \boldsymbol{y}) \prod_{\boldsymbol{x} \in \mathcal{U}} p_\theta(\boldsymbol{x}),$$

To derive a lower bound for the overall objective, we need to obtain lower bounds on $\log p_\theta(\boldsymbol{x})$ and $\log p_\theta(\boldsymbol{x}, \boldsymbol{y})$. When the labels are unobserved the latent state will consist of $\boldsymbol{z}$ and $\boldsymbol{y}$. Using the

factorization according to the graph in Figure 2 yields

$$\log p_\theta(\boldsymbol{x}) \geq E_{q_\phi(\boldsymbol{z}|\boldsymbol{x})q_\varphi(\boldsymbol{y}|\boldsymbol{z}_c)} \left[ \log \left( \frac{p_\theta(\boldsymbol{x} \mid \boldsymbol{z})p_\psi(\boldsymbol{z} \mid \boldsymbol{y})p(\boldsymbol{y})}{q_\varphi(\boldsymbol{y} \mid \boldsymbol{z}_c)q_\phi(\boldsymbol{z} \mid \boldsymbol{x})} \right) \right],$$

where $p_\psi(\boldsymbol{z} \mid \boldsymbol{y}) = p(\boldsymbol{z}_{\backslash c})p_\psi(\boldsymbol{z}_c \mid \boldsymbol{y})$. For supervised data points we consider a lower bound on the likelihood $p_\theta(\boldsymbol{x}, \boldsymbol{y})$,

$$\log p_\theta(\boldsymbol{x}, \boldsymbol{y}) \geq \int \log \frac{p_\theta(\boldsymbol{x} \mid \boldsymbol{z})p_\psi(\boldsymbol{z} \mid \boldsymbol{y})p(\boldsymbol{y})}{q_{\varphi,\phi}(\boldsymbol{z} \mid \boldsymbol{x}, \boldsymbol{y})} q_{\varphi,\phi}(\boldsymbol{z} \mid \boldsymbol{x}, \boldsymbol{y})\mathrm{d}\boldsymbol{z},$$

in order to make sense of the term $q_{\varphi,\phi}(\boldsymbol{z} \mid \boldsymbol{x}, \boldsymbol{y})$, which is usually different from $q_\phi(\boldsymbol{z} \mid \boldsymbol{x})$ we consider the inference model

$$q_{\varphi,\phi}(\boldsymbol{z} \mid \boldsymbol{x}, \boldsymbol{y}) = \frac{q_\varphi(\boldsymbol{y} \mid \boldsymbol{z}_c)q_\phi(\boldsymbol{z} \mid \boldsymbol{x})}{q_{\varphi,\phi}(\boldsymbol{y} \mid \boldsymbol{x})}, \quad \text{where} \quad q_{\varphi,\phi}(\boldsymbol{y} \mid \boldsymbol{x}) = \int q_\varphi(\boldsymbol{y} \mid \boldsymbol{z}_c)q_\phi(\boldsymbol{z} \mid \boldsymbol{x})\mathrm{d}\boldsymbol{z}.$$

Returning to the lower bound on $\log p_\theta(\boldsymbol{x}, \boldsymbol{y})$ we obtain

$$\begin{aligned}
\log p_\theta(\boldsymbol{x}, \boldsymbol{y}) &\geq \int \log \frac{p_\theta(\boldsymbol{x} \mid \boldsymbol{z})p_\psi(\boldsymbol{z} \mid \boldsymbol{y})p(\boldsymbol{y})}{q(\boldsymbol{z} \mid \boldsymbol{x}, \boldsymbol{y})} q(\boldsymbol{z} \mid \boldsymbol{x}, \boldsymbol{y})\mathrm{d}\boldsymbol{z} \\
&= \int \log \left( \frac{p_\theta(\boldsymbol{x} \mid \boldsymbol{z})p_\psi(\boldsymbol{z} \mid \boldsymbol{y})p(\boldsymbol{y})q_{\varphi,\phi}(\boldsymbol{y} \mid \boldsymbol{x})}{q_\varphi(\boldsymbol{y} \mid \boldsymbol{z}_c)q_\phi(\boldsymbol{z} \mid \boldsymbol{x})} \right) \frac{q_\varphi(\boldsymbol{y} \mid \boldsymbol{z}_c)q_\phi(\boldsymbol{z} \mid \boldsymbol{x})}{q_{\varphi,\phi}(\boldsymbol{y} \mid \boldsymbol{x})}\mathrm{d}\boldsymbol{z} \\
&= E_{q_\phi(\boldsymbol{z}|\boldsymbol{x})} \left[ \frac{q_\varphi(\boldsymbol{y} \mid \boldsymbol{z}_c)}{q_{\varphi,\phi}(\boldsymbol{y} \mid \boldsymbol{x})} \log \left( \frac{p(\boldsymbol{x} \mid \boldsymbol{z})p_\psi(\boldsymbol{z}_c \mid \boldsymbol{y})}{q_\varphi(\boldsymbol{y} \mid \boldsymbol{z}_c)q_\phi(\boldsymbol{z} \mid \boldsymbol{x})} \right) \right] + \log q_{\varphi,\phi}(\boldsymbol{y} \mid \boldsymbol{x}) + \log p(\boldsymbol{y}),
\end{aligned}$$

where $q_\varphi(\boldsymbol{y} \mid \boldsymbol{z}_c)/q_{\varphi,\phi}(\boldsymbol{y} \mid \boldsymbol{x})$ denotes the Radon-Nikodym derivative of $q_{\varphi,\phi}(\boldsymbol{z} \mid \boldsymbol{x}, \boldsymbol{y})$ with respect to $q_\phi(\boldsymbol{z} \mid \boldsymbol{x})$.

### B.2 Alternative Derivation of Unsupervised Bound

The bound for the unsupervised case can alternatively be derived by applying Jensen's inequality twice. First, use the standard (unsupervised) ELBO

$$\log p_\theta(\boldsymbol{x}) \geq \mathbb{E}_{q_\phi(\boldsymbol{z}|\boldsymbol{x})} \left[ \log \frac{p_\theta(\boldsymbol{x} \mid \boldsymbol{z})p(\boldsymbol{z})}{q_\phi(\boldsymbol{z} \mid \boldsymbol{x})} \right].$$

Now, since calculating $p(\boldsymbol{z}) = p(\boldsymbol{z}_c)p(\boldsymbol{z}_{\backslash c}) = p(\boldsymbol{z}_{\backslash c}) \sum_{\boldsymbol{y}} p(\boldsymbol{z}_c \mid \boldsymbol{y})p(\boldsymbol{y})$ can be expensive we can apply Jensen's inequality a second time to the expectation over $\boldsymbol{z}_c$ to obtain

$$\log p(\boldsymbol{z}_c) \geq \mathbb{E}_{q_\varphi(\boldsymbol{y}|\boldsymbol{z}_c)} \left[ \log \frac{p_\psi(\boldsymbol{z}_s \mid \boldsymbol{y})p(\boldsymbol{y})}{q_\varphi(\boldsymbol{y} \mid \boldsymbol{z}_s)} \right].$$

Substituting this bound into the unsupervised ELBO yields again our bound

$$\log p(\boldsymbol{x}) \geq \mathbb{E}_{q_\phi(\boldsymbol{z}|\boldsymbol{x})q_\varphi(\boldsymbol{y}|\boldsymbol{z}_c)} \left[ \log \frac{p_\theta(\boldsymbol{x} \mid \boldsymbol{z})p(\boldsymbol{z} \mid \boldsymbol{y})}{q_\phi(\boldsymbol{z} \mid \boldsymbol{x})q_\varphi(\boldsymbol{y} \mid \boldsymbol{z}_c)} \right] + \log p(\boldsymbol{y}) \tag{7}$$

## C Implementation

### C.1 CelebA

We chose to use only a subset of the labels present in CelebA, since not all attributes are visually distinguishable in the reconstructions e.g. (earrings). As such we limited ourselves to the following labels: arched eyebrows, bags under eyes, bangs, black hair, blond hair, brown hair, bushy eyebrows, chubby, eyeglasses, heavy makeup, male, no beard, pale skin, receding hairline, smiling, wavy hair, wearing necktie, young. No images were omitted or cropped, the only modifications were keeping the aforementioned labels and resizing the images to be $64 \times 64$ in dimension.

### C.2 Chexpert

The Chexpert dataset comprises of chest X-rays taken from a variety of patients. We down-sampled each image to be $64 \times 64$ and used the same networks from the CelebA experiments. The five main attributes for Chexpert are: cardiomegaly, edema, consolidation, atelectasis, pleural effusion. Which for non medical experts can be interpreted as: enlargement of the heart; fluid in the alveoli; fluid in the lungs; collapsed lung; fluid in the corners of the lungs.

### C.3 IMPLEMENTATION DETAILS

For our experiments we define the generative and inference networks as follows. The approximate posterior is represented as $q_\phi(\boldsymbol{z} \mid \boldsymbol{x}) = \mathcal{N}(\boldsymbol{z_c}, \boldsymbol{z_{\backslash c}} \mid \boldsymbol{\mu}_\phi(\boldsymbol{x}), \text{diag}(\boldsymbol{\sigma}_\phi^2(\boldsymbol{x})))$ with $\boldsymbol{\mu}_\phi(\boldsymbol{x})$ and $\text{diag}(\boldsymbol{\sigma}_\phi^2(\boldsymbol{x}))$ being the architecture from Higgins et al. (2016). The generative model $p_\theta(\boldsymbol{x} \mid \boldsymbol{z})$ is represented by a Laplace distribution, again parametrized using the architecture from Higgins et al. (2016). The label predictive distribution $q_\varphi(\boldsymbol{y} \mid \boldsymbol{z}_c)$ is represented as $\text{Ber}(\boldsymbol{y} \mid \boldsymbol{\pi}_\varphi(\boldsymbol{z_c}))$ with $\boldsymbol{\pi}_\varphi(\boldsymbol{z_c})$ being a diagonal transformation forcing the factorisation $q_\varphi(\boldsymbol{y} \mid \boldsymbol{z}_c) = \prod_i q_{\psi^i}(y_i \mid \boldsymbol{z_c}^i)$. The conditional prior is given as $p_\psi(\boldsymbol{z_c} \mid \boldsymbol{y}) = \mathcal{N}(\boldsymbol{z_c} \mid \boldsymbol{\mu}_\psi(\boldsymbol{y}), \text{diag}(\boldsymbol{\sigma}_\psi^2(\boldsymbol{y})))$, with the appropriate factorisation, where the parameters are represented by an MLP. Finally, the prior placed on the portion of the latent space reserved for unlabelled latent variables is $p(\boldsymbol{z_{\backslash c}}) = \mathcal{N}(\boldsymbol{z_{\backslash c}} \mid \boldsymbol{0}, \mathbf{I}))$. For the latent space $\boldsymbol{z_c} \in \mathbb{R}^{m_c}$ and $\boldsymbol{z_{\backslash c}} \in \mathbb{R}^{m_{\backslash c}}$, where $m = m_c + m_{\backslash c}$ with $m_c = 18$ and $m_{\backslash c} = 27$ for CelebA. The architectures are given in and Table 3.

| Encoder | Decoder |
|---|---|
| Input 32 x 32 x 3 channel image | Input $\in \mathbb{R}^m$ |
| $32 \times 3 \times 4 \times 4$ Conv2d stride 2 & ReLU | $m \times 256$ Linear layer |
| $32 \times 32 \times 4 \times 4$ Conv2d stride 2 & ReLU | $128 \times 256 \times 4 \times 4$ ConvTranspose2d stride 1 & ReLU |
| $64 \times 32 \times 4 \times 4$ Conv2d stride 2 & ReLU | $64 \times 128 \times 4 \times 4$ ConvTranspose2d stride 2 & ReLU |
| $128 \times 64 \times 4 \times 4$ Conv2d stride 2 & ReLU | $32 \times 64 \times 4 \times 4$ ConvTranspose2d stride 2 & ReLU |
| $256 \times 128 \times 4 \times 4$ Conv2d stride 1 & ReLU | $32 \times 32 \times 4 \times 4$ ConvTranspose2d stride 2 & ReLU |
| $256 \times (2 \times m)$ Linear layer | $3 \times 32 \times 4 \times 4$ ConvTranspose2d stride 2 & Sigmoid |

| Classifier | Conditional Prior |
|---|---|
| Input $\in \mathbb{R}^{m_c}$ | Input $\in \mathbb{R}^{m_c}$ |
| $m_c \times m_c$ Diagonal layer | $m_c \times m_c$ Diagonal layer |

Table 3: Architectures for CelebA and Chexpert.

**Optimization** We trained the models on a GeForce GTX Titan GPU. Training consumed $\sim 2\text{Gb}$ for CelebA and Chexpert, taking around 2 hours to complete 100 epochs respectively. Both models were optimized using Adam with a learning rate of $2 \times 10^{-4}$ for CelebA respectively.

#### C.3.1 HIGH VARIANCE OF CLASSIFIER GRADIENTS

The gradients of the classifier parameters $\varphi$ suffer from a high variance during training. We find that not reparameterizing $\boldsymbol{z_c}$ for $q_\varphi(\boldsymbol{y} \mid \boldsymbol{z}_c)$ reduces this issue:

$$\mathcal{L}_{\text{CCVAE}}(\boldsymbol{x}, \boldsymbol{y}) = \mathbb{E}_{q_\phi(\boldsymbol{z}\mid\boldsymbol{x})}\left[\frac{q_\varphi(\boldsymbol{y} \mid \bar{\boldsymbol{z}}_{\boldsymbol{c}})}{q_{\varphi,\phi}(\boldsymbol{y} \mid \boldsymbol{x})} \log \frac{p_\theta(\boldsymbol{x} \mid \boldsymbol{z})p_\psi(\boldsymbol{z} \mid \boldsymbol{y})}{q_\varphi(\boldsymbol{y} \mid \bar{\boldsymbol{z}}_{\boldsymbol{c}})q_\phi(\boldsymbol{z} \mid \boldsymbol{x})}\right] + \log q_{\varphi,\phi}(\boldsymbol{y} \mid \boldsymbol{x}) + \log p(\boldsymbol{y}). \quad (8)$$

where $\bar{\boldsymbol{z}}_{\boldsymbol{c}}$ indicates that we do not reparameterize the sample. This significantly reduces the variance of the magnitude of the gradient norm $\nabla_\varphi$, allowing the classifier to learn appropriate weights and structure the latent space. This can be seen in Figure 8, where we plot the gradient norm of $\varphi$ for when we **do** reparameterize $\boldsymbol{z_c}$ (blue) and when we **do not** (orange). Clearly not reparameterizing leads to a lower variance in the gradient norm of the classifier, which aides learning. To a certain extent these gradients can be viewed as redundant, as there is already gradients to update the predictive distribution due to the $\log q_{\varphi,\phi}(\boldsymbol{y} \mid \boldsymbol{x})$ term anyway.

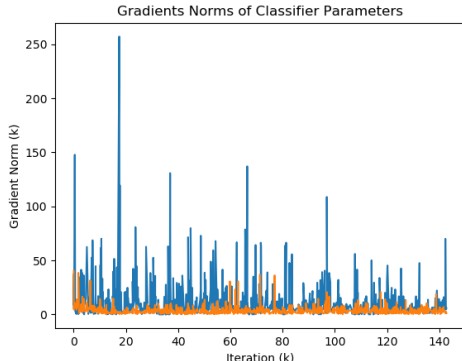

Figure 8: Gradient norms of classifier.

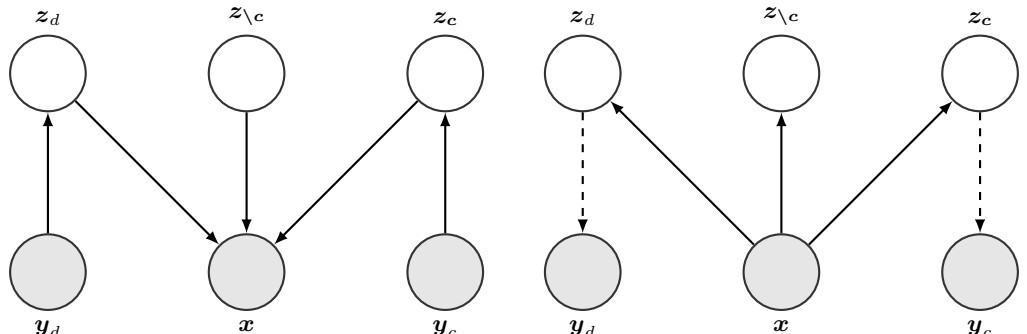

Figure 9: Left: Generative model for DIVA, Right: Inference model where dashed line indicates auxiliary classifier.

## C.4 MODIFIED DIVA

The primary goal of DIVA is domain invariant classification and not to obtain representations of individual characteristics like we do here. The objective is essentially a classifier which is regularized by a variational objective. However, to achieve domain generalization, the authors aim to disentangle the domain, class and other generative factors. This motivation leads to a graphical model that is similar in spirit to ours ( Figure 9), in that the latent variables are used to predict labels, and the introduction of the inductive bias to partition the latent space. As such, DIVA can be modified to suit our problem of encapsulating characteristics. The first modification we need to consider is the removal of $z_d$, as we are not considering multi-domain problems. Secondly, we introduce the factorization present in CCVAE, namely $q_\varphi(\boldsymbol{y} \mid \boldsymbol{z}_c) = \prod_i q_{\psi^i}(y_i | \boldsymbol{z_c}^i)$. With these two modifications an alternative objective can now be constructed, with the supervised given as

$$\mathcal{L}_{SDIVA}(\boldsymbol{x}, \boldsymbol{y}) = \mathbb{E}_{q_\phi(\boldsymbol{z}|\boldsymbol{x})} \log p_\theta(\boldsymbol{x} \mid \boldsymbol{z}) - \beta KL(q_\phi(\boldsymbol{z}_{\backslash \boldsymbol{c}}|x)||p(\boldsymbol{z}_{\backslash \boldsymbol{c}}))$$
$$- \beta KL(q_\phi(\boldsymbol{z_c}|x)||p_\psi(\boldsymbol{z}_c \mid \boldsymbol{y})),$$

and the unsupervised as

$$\mathcal{L}_{UDIVA}(\boldsymbol{x}) = \mathbb{E}_{q_\phi(\boldsymbol{z}|\boldsymbol{x})} \log p_\theta(\boldsymbol{x} \mid \boldsymbol{z}) - \beta KL(q_\phi(\boldsymbol{z}_{\backslash \boldsymbol{c}}|x)||p(\boldsymbol{z}_{\backslash \boldsymbol{c}}))$$
$$+ \beta \mathbb{E}_{q_\phi(\boldsymbol{z_c}|x)q_\varphi(\boldsymbol{y}|\boldsymbol{z}_c)}[\log p_\psi(\boldsymbol{z}_c \mid \boldsymbol{y}) - \log q_\phi(\boldsymbol{z_c}|x)],$$
$$+ \beta \mathbb{E}_{q_\phi(\boldsymbol{z_c}|x)q_\varphi(\boldsymbol{y}|\boldsymbol{z}_c)}[\log p(\boldsymbol{y}) - \log q_\varphi(\boldsymbol{y} \mid \boldsymbol{z}_c)],$$

where $\boldsymbol{y}$ has to be imputed. The final objective for DIVA is then given as

$$\log p_\theta\left(\mathcal{D}\right) \geq \sum_{(\boldsymbol{x}, \boldsymbol{y}) \in \mathcal{S}} \mathcal{L}_{SDIVA}(\boldsymbol{x}, \boldsymbol{y}) + \sum_{\boldsymbol{x} \in \mathcal{U}} \left[\mathcal{L}_{UDIVA}(\boldsymbol{x}) + \alpha \mathbb{E}_{q(\boldsymbol{z_c}|\boldsymbol{x})} \log q_\varphi(\boldsymbol{y} \mid \boldsymbol{z}_c)\right].$$

It is interesting to note the differences to the objective of CCVAE, namely, there is no emergence of a natural classifier in the supervised case, and $\boldsymbol{y}$ has to be imputed in the unsupervised case instead of relying on variational inference as in CCVAE. Clearly such differences have a significant impact on performance as demonstrated by the main results of this paper.

## D ADDITIONAL RESULTS

### D.1 SINGLE INTERVENTIONS

Here we demonstrate single interventions where we change the binary value for the desired attributes. To quantitatively evaluate the single interventions, we intervene on a single label and report the changes in log-probabilities assigned by a pre-trained classifier. If the single intervention only affects the characteristics of the chosen label, then there should be no change in other classes and only a change on the chosen label. Intervening on all possible labels yields a confusion matrix, with the optimal results being a diagonal matrix with zero off-diagonal elements. We also report the condition number for the confusion matrices, given in the titles.

It is interesting to note that the interventions for CCVAE are subtle, this is due to the latent $z_c^i \sim p(z_c^i|y_i)$, which will be centered around the mean. More striking intervention can be achieved by traversing along $z_c^i$.

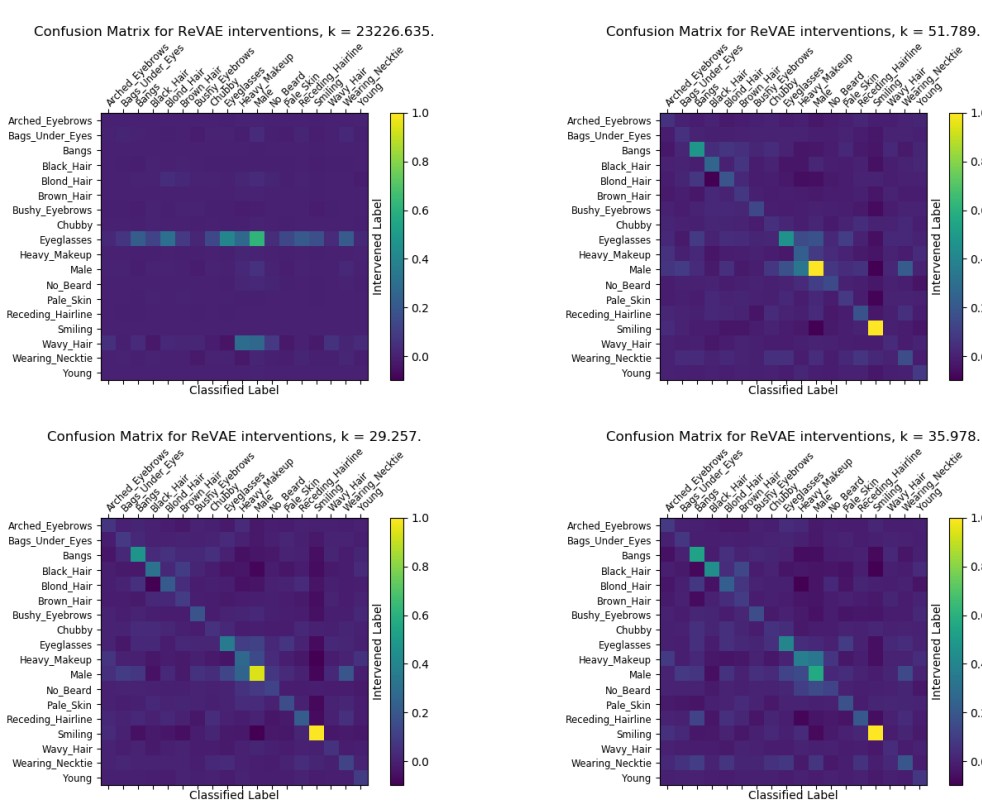

Figure 10: Confusion matrices for CCVAE for (from top left clockwise) $f = 0.004, 0.06, 0.2, 1.0$

$f = 0.004$

$f = 0.06$

$f = 0.2$

$f = 1.0$

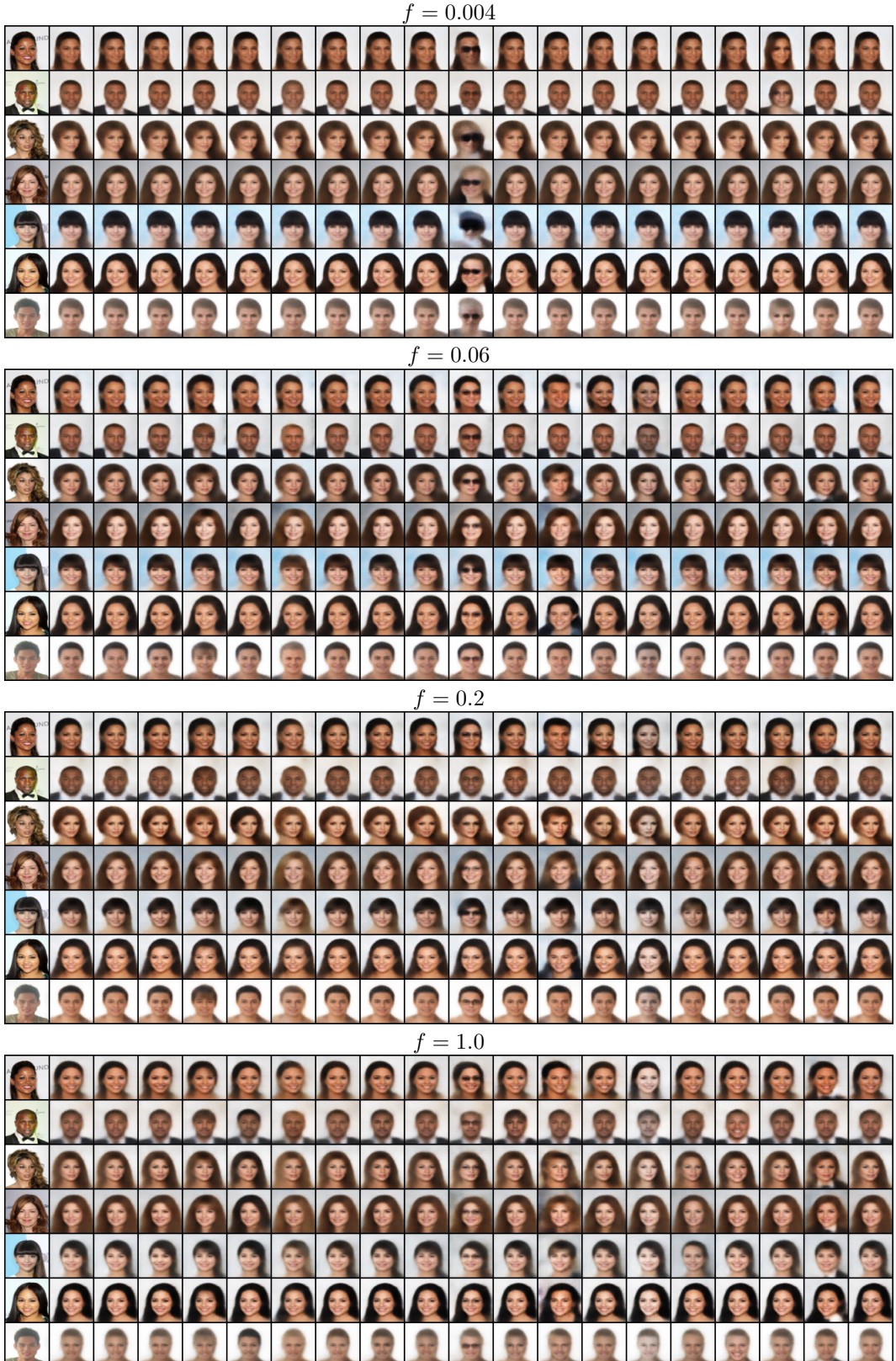

Figure 11: CCVAE. From left to right: original, reconstruction, then interventions from switching on the following labels: `arched eyebrows`, `bags under eyes`, `bangs`, `black hair`, `blond hair`, `brown hair`, `bushy eyebrows`, `chubby`, `eyeglasses`, `heavy makeup`, `male`, `no beard`, `pale skin`, `receding hairline`, `smiling`, `wavy hair`, `wearing necktie`, `young`.

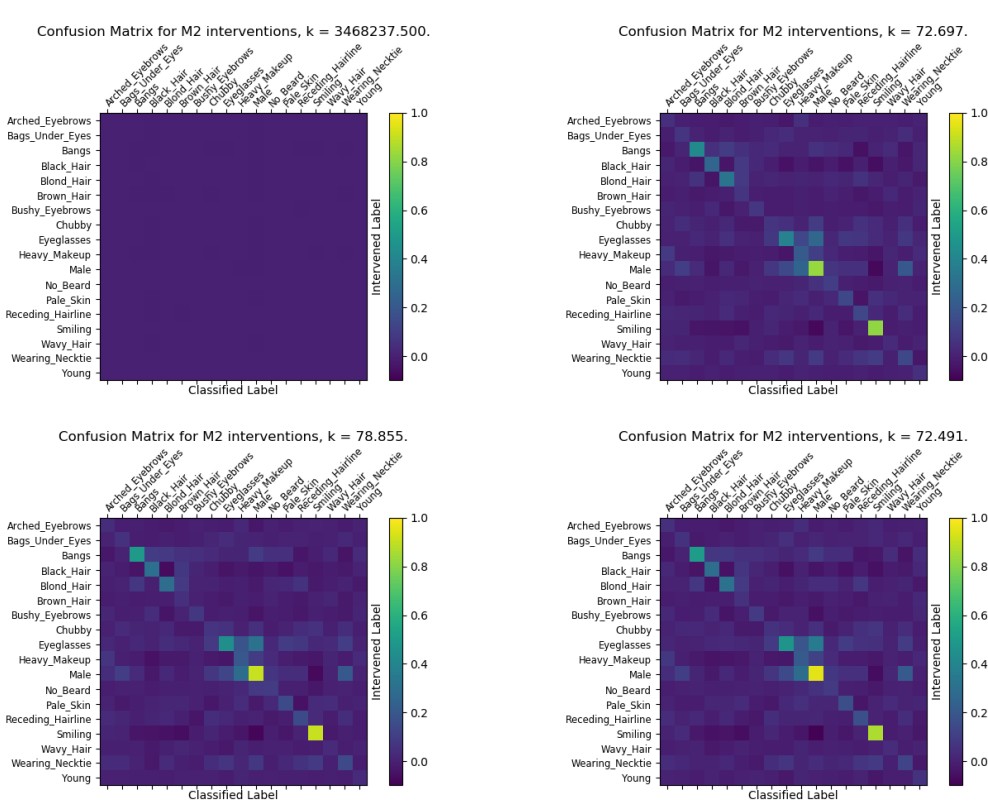

Figure 12: Confusion matrices for M2 for (from top left clockwise) $f = 0.004, 0.06, 0.2, 1.0$

$f = 0.004$

$f = 0.06$

$f = 0.2$

$f = 1.0$

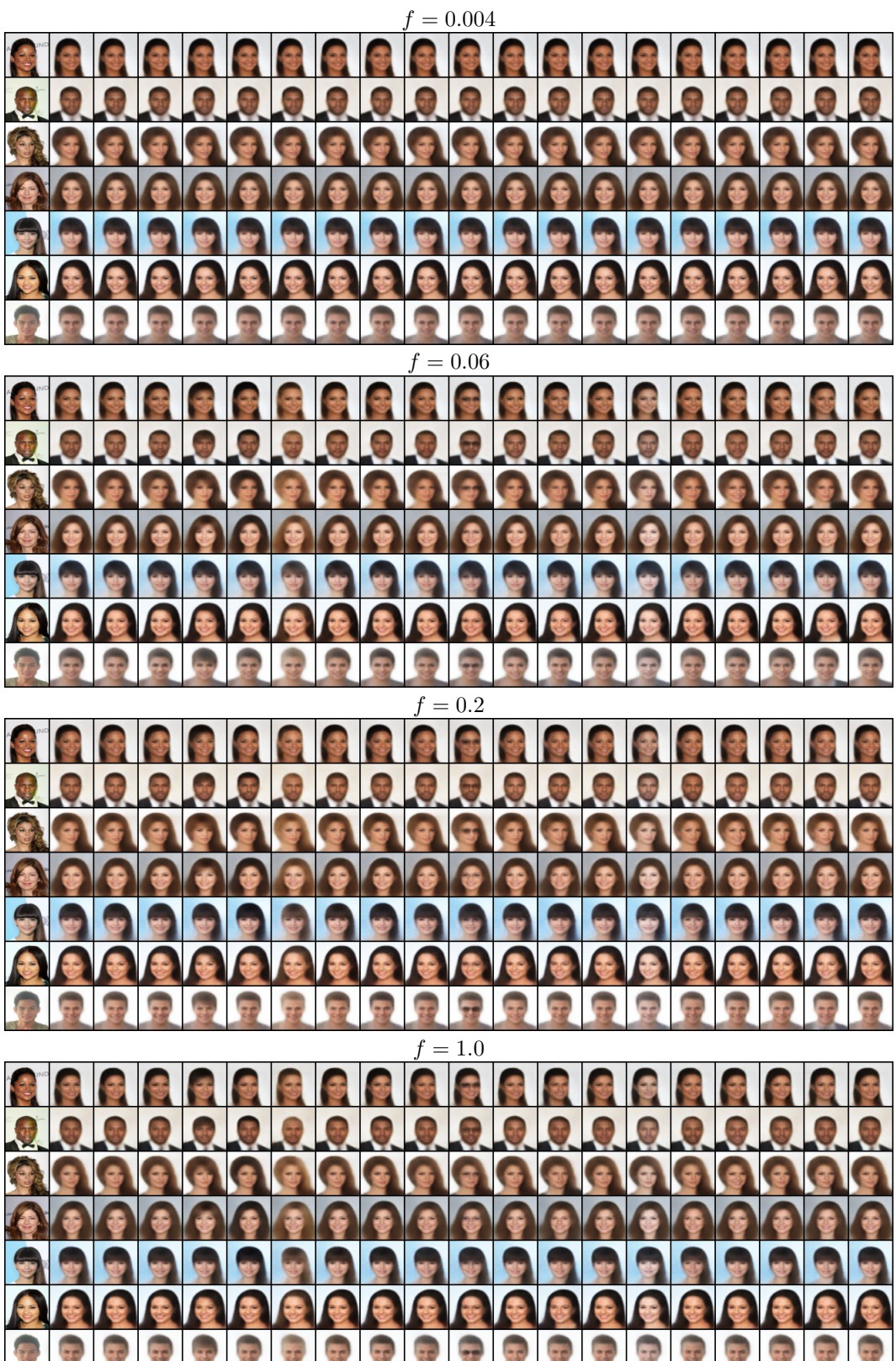

Figure 13: M2. From left to right: original, reconstruction, then interventions from switching on the following labels: `arched eyebrows`, `bags under eyes`, `bangs`, `black hair`, `blond hair`, `brown hair`, `bushy eyebrows`, `chubby`, `eyeglasses`, `heavy makeup`, `male`, `no beard`, `pale skin`, `receding hairline`, `smiling`, `wavy hair`, `wearing necktie`, `young`.

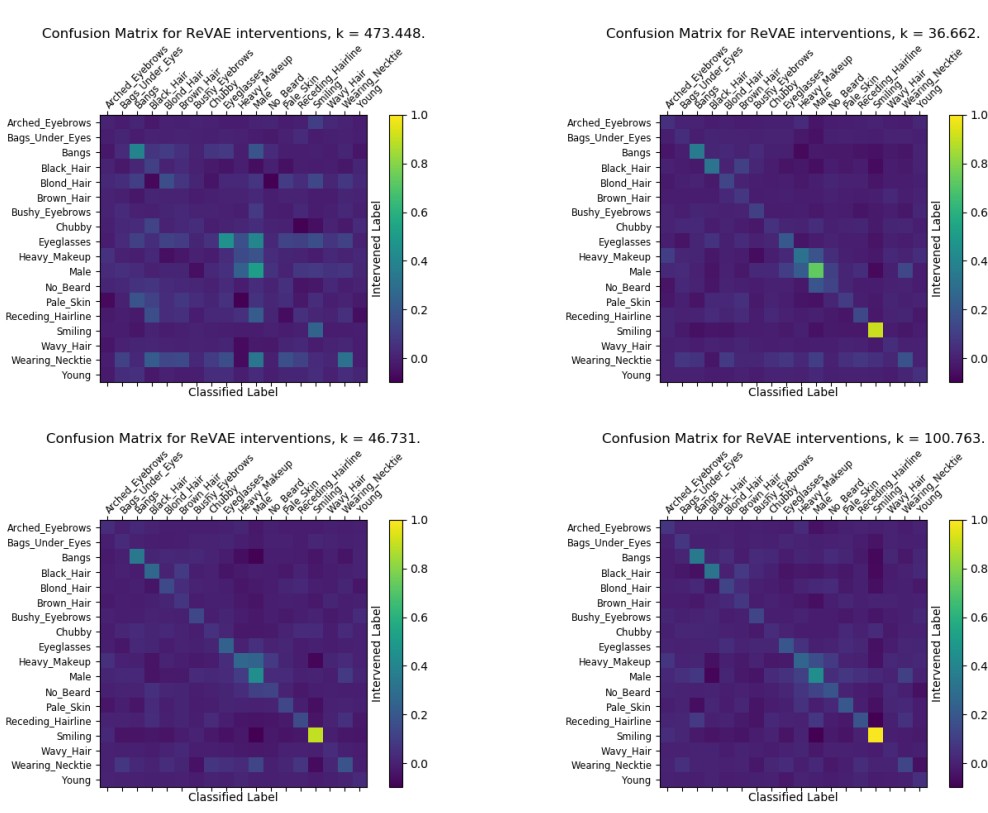

Figure 14: Confusion matrices for DIVA for (from top left clockwise) $f = 0.004, 0.06, 0.2, 1.0$

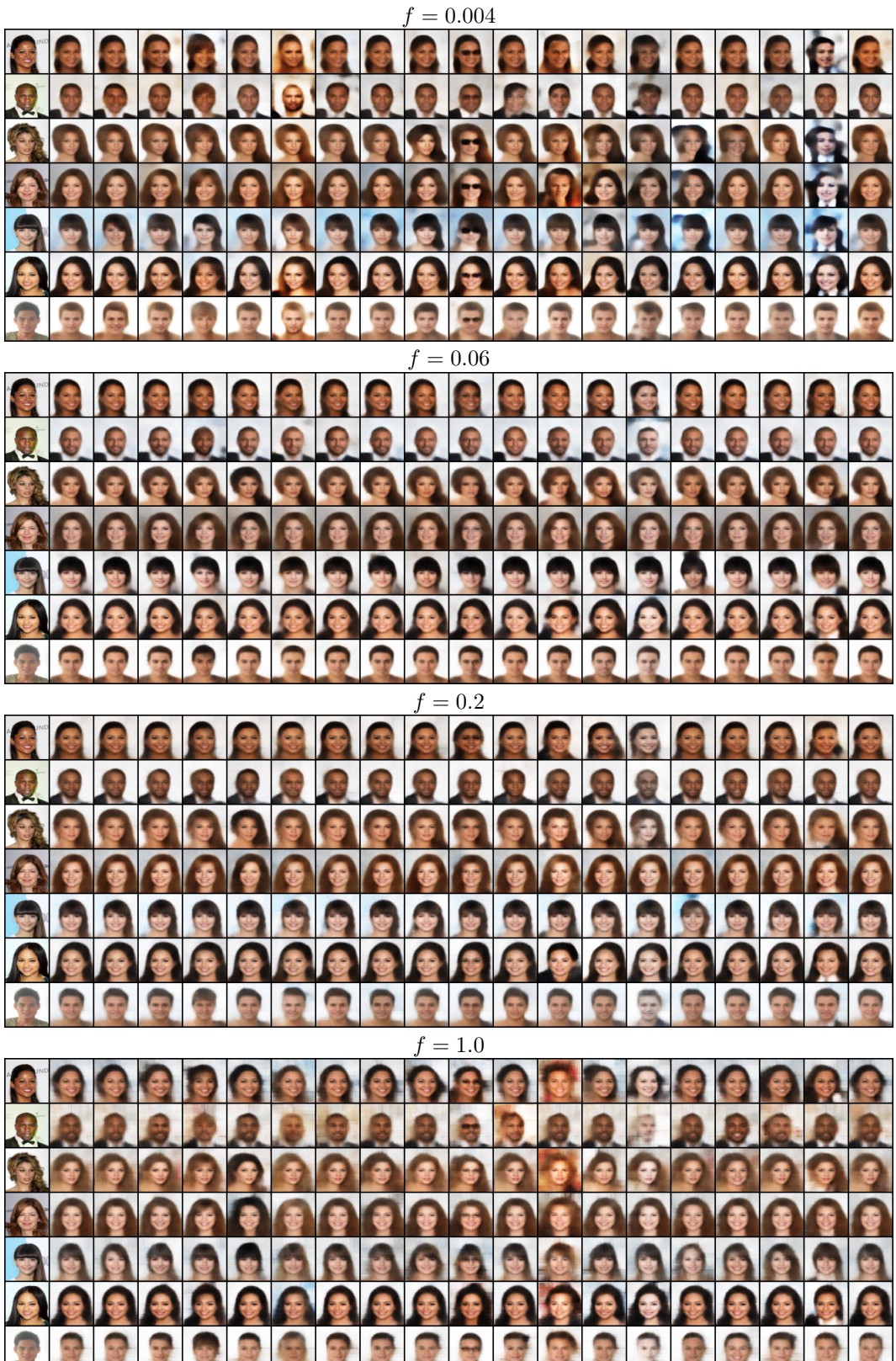

Figure 15: DIVA. From left to right: original, reconstruction, then interventions from switching on the following labels: `arched eyebrows`, `bags under eyes`, `bangs`, `black hair`, `blond hair`, `brown hair`, `bushy eyebrows`, `chubby`, `eyeglasses`, `heavy makeup`, `male`, `no beard`, `pale skin`, `receding hairline`, `smiling`, `wavy hair`, `wearing necktie`, `young`.

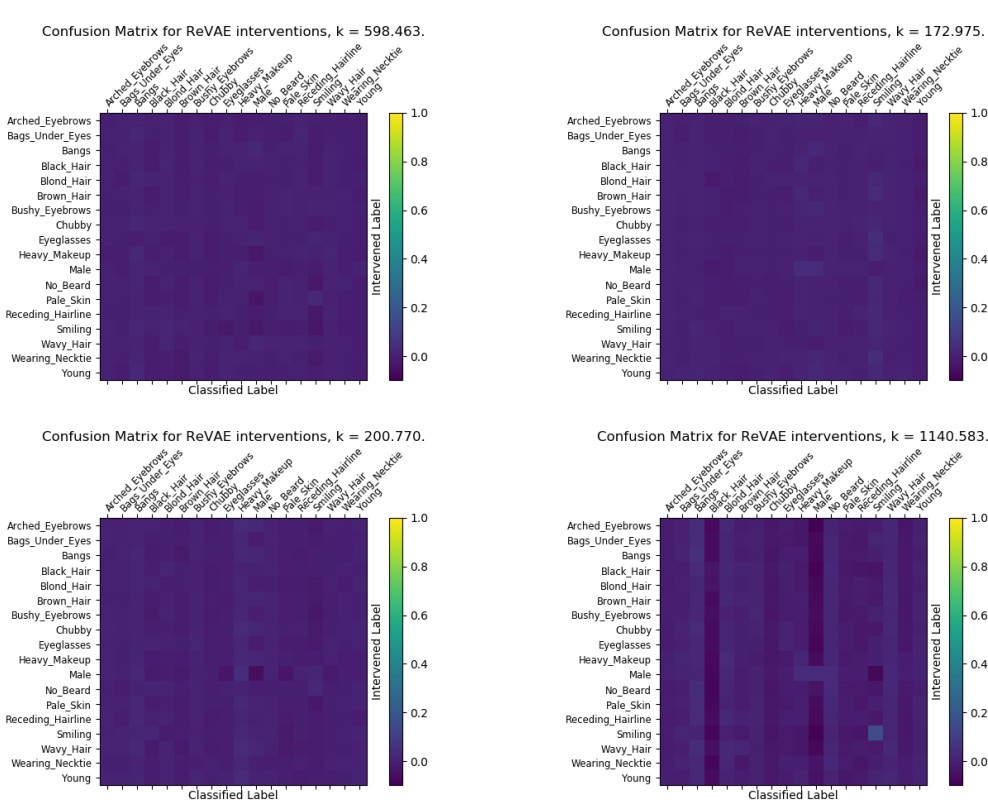

Figure 16: Confusion matrices for MVAE for (from top left clockwise) $f = 0.004, 0.06, 0.2, 1.0$

$f = 0.004$

$f = 0.06$

$f = 0.2$

$f = 1.0$

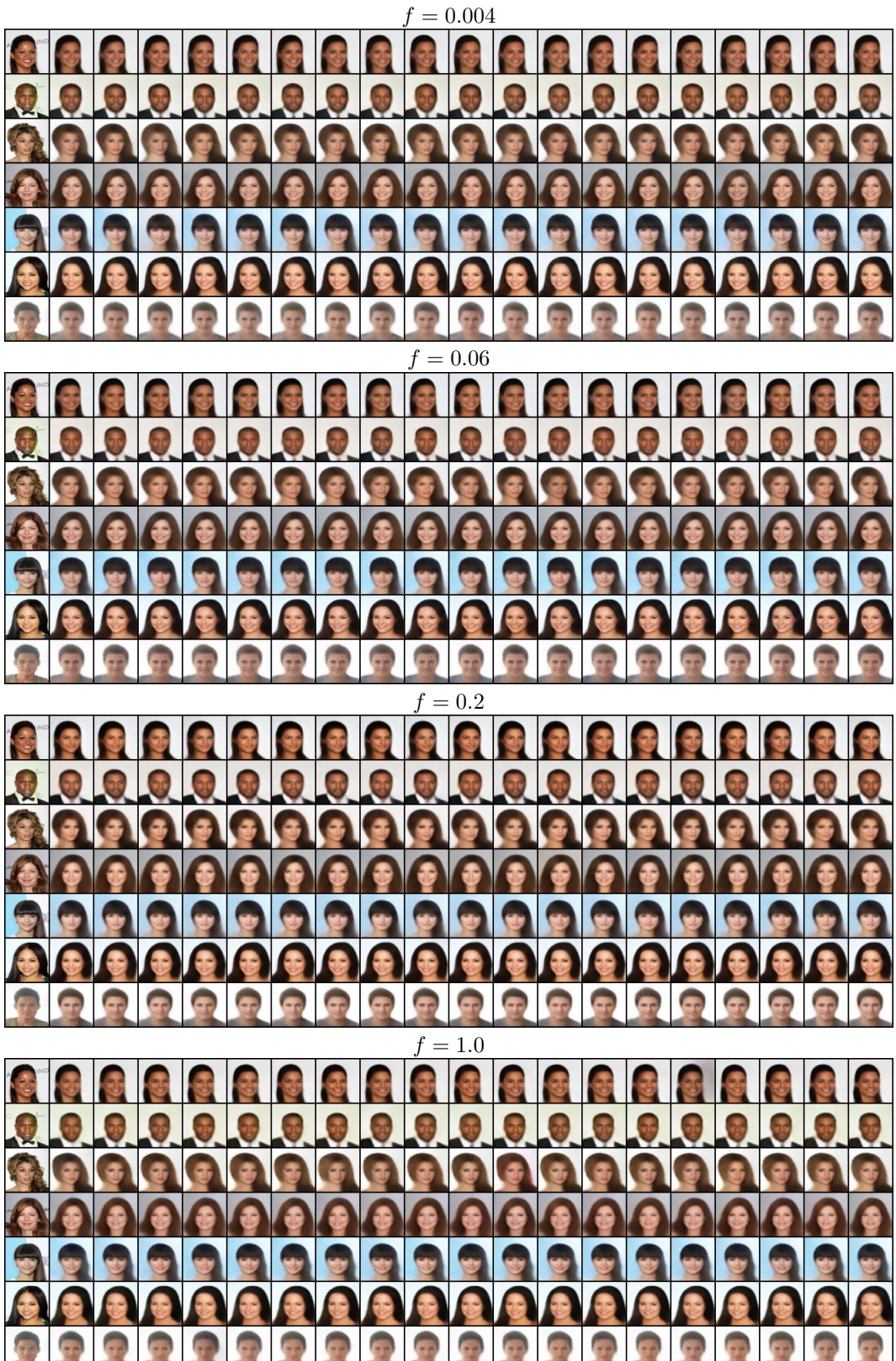

Figure 17: MVAE. From left to right: original, reconstruction, then interventions from switching on the following labels: arched eyebrows, bags under eyes, bangs, black hair, blond hair, brown hair, bushy eyebrows, chubby, eyeglasses, heavy makeup, male, no beard, pale skin, receding hairline, smiling, wavy hair, wearing necktie, young.

## D.2 Latent Traversals

Here we provide more latent traversals for CCVAE in Figure 18 and for DIVA in Figure 19. CCVAE is able to smoothly alter characteristics, indicating that it is able to encapsulate characteristics in a single dimension, unlike DIVA which is unable to alter the characteristics effectively, suggesting it cannot encapsulate the characteristics.

## D.3 Generation

We provide results for the fidelity of image generation on CelebA. To do this we use the FID metric Heusel et al. (2017), we omitted results for Chexpert as the inception model used in FID has not been trained on the typical features associated with X-Rays. The results are given in Table 4, interestingly for low supervision rates MVAE obtains the best performance but for higher supervision rates M2 outperforms MVAE. We posit that this is due to MVAE having little structure imposed on the latent space, as such the POE can structure the representation purely for reconstruction without considering the labels, something which is not possible as the supervision rate is increased. CCVAE obtains competitive results with respect to M2. It is important to note that generative fidelity is not the focus of this work as we focus purely on how to structure the latent space using labels. It is no surprise then that the generations are bad as structuring the latent space will potentially be at odds with the reconstruction term in the loss.

Table 4: CelebA FID scores.

| Model | $f = 0.004$ | $f = 0.06$ | $f = 0.2$ | $f = 1.0$ |
|---|---|---|---|---|
| CCVAE | 127.956 | 121.84 | 121.751 | 120.457 |
| M2 | 127.719 | 122.521 | **120.406** | **119.228** |
| DIVA | 192.448 | 230.522 | 218.774 | 201.484 |
| MVAE | **118.308** | **115.947** | 128.867 | 137.461 |

## D.4 Conditional Generation

To asses conditional generation, we first train an independent classifier for both datasets. We then conditionally generate samples given labels and evaluate them using this pre-trained classifier. Results provided in Table 5. CCVAE and M2 are comparable in generative abilities, but DIVA and MVAE perform poorly, indicated by random guessing.

Table 5: Generations accuracies.

| Model | CelebA | | | | Chexpert | | | |
|---|---|---|---|---|---|---|---|---|
| | $f = 0.004$ | $f = 0.06$ | $f = 0.2$ | $f = 1.0$ | $f = 0.004$ | $f = 0.06$ | $f = 0.2$ | $f = 1.0$ |
| CCVAE | **0.513** | 0.605 | **0.612** | 0.596 | **0.516** | **0.563** | **0.549** | 0.542 |
| M2 | 0.499 | **0.61** | 0.612 | **0.611** | 0.503 | 0.547 | 0.547 | **0.558** |
| DIVA | 0.501 | 0.501 | 0.501 | 0.501 | 0.499 | 0.503 | 0.503 | 0.503 |
| MVAE | 0.501 | 0.501 | 0.501 | 0.501 | 0.499 | 0.499 | 0.499 | 0.499 |

## D.5 Diversity of Conditional Generations

We also report more examples for diversity, as in Figure 5, in Figure 20.

## D.6 Multi-class Setting

Here we provide results for the multi-class setting of MNIST and FashionMNIST. The multi-class setting is somewhat tangential to our work, but we include it for completeness. For CCVAE, we have some flexibility over the size of the latent space. Trying to encapsulate representations for each label is not well suited for this setting, as it's not clear how you could alter the representation of an image being a 6, whilst preserving the representation of it being an 8. In fact, there is really only one label for this setting, but it takes multiple values. With this in mind, we can now make an explicit choice about how the latent space will be structured, we can set $z_c \in \mathbb{R}$ or $z_c \in \mathbb{R}^N$, or conversely, store all of the representation in $z_c$, i.e. $z_{\setminus c} = \emptyset$. Furthermore, we do not need to enforce the factorization $q_\varphi(y \mid z_c) = \prod_i q(y_i|z_c^i)$, and instead can be parameterized by a function $\mathcal{F} : \mathbb{R}^N \to \mathbb{R}^M$ where M is the number of possible classes.

**Classification** We provide the classification results in Table 6.

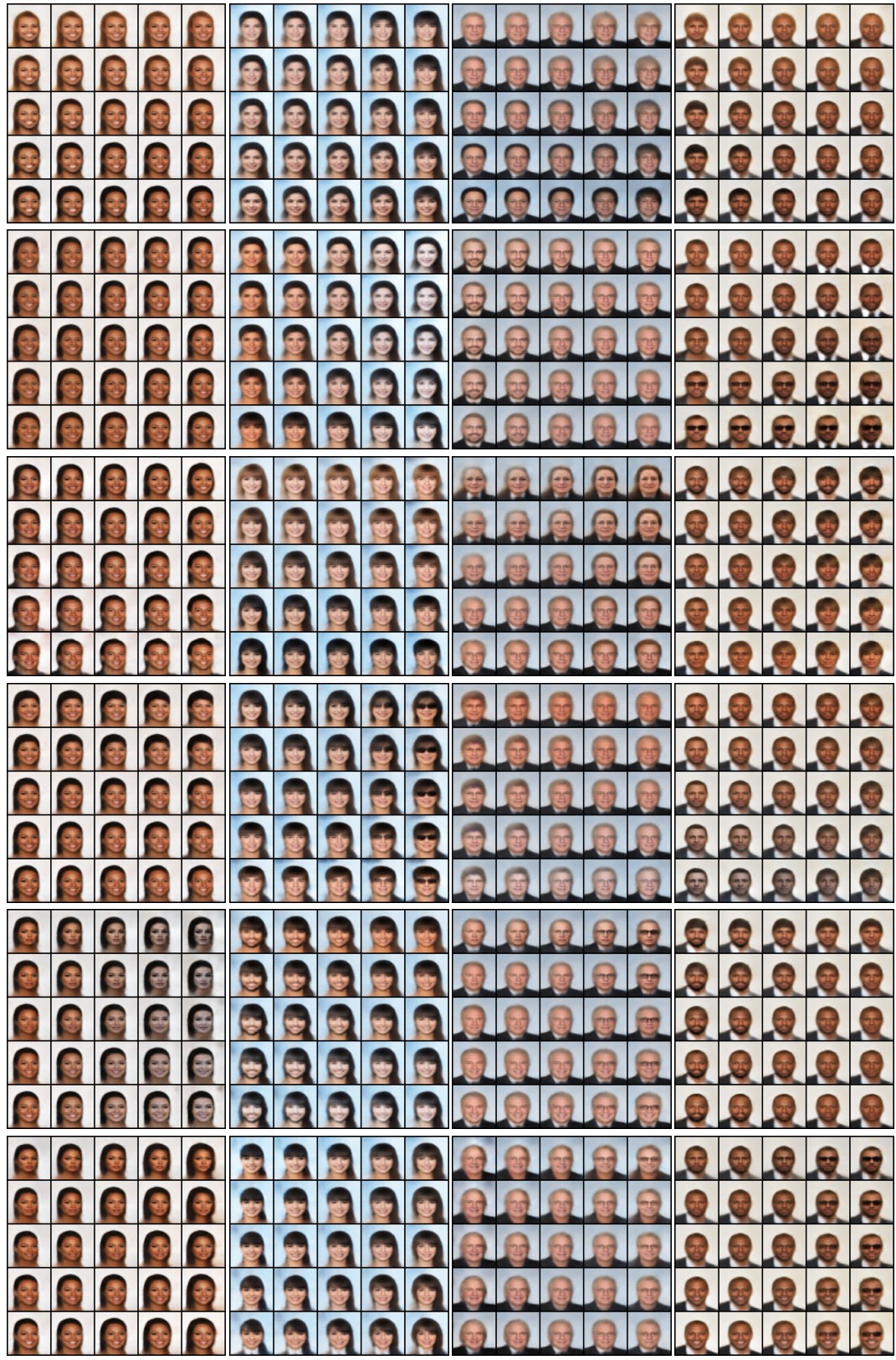

Figure 18: Various latent traversals for CCVAE.

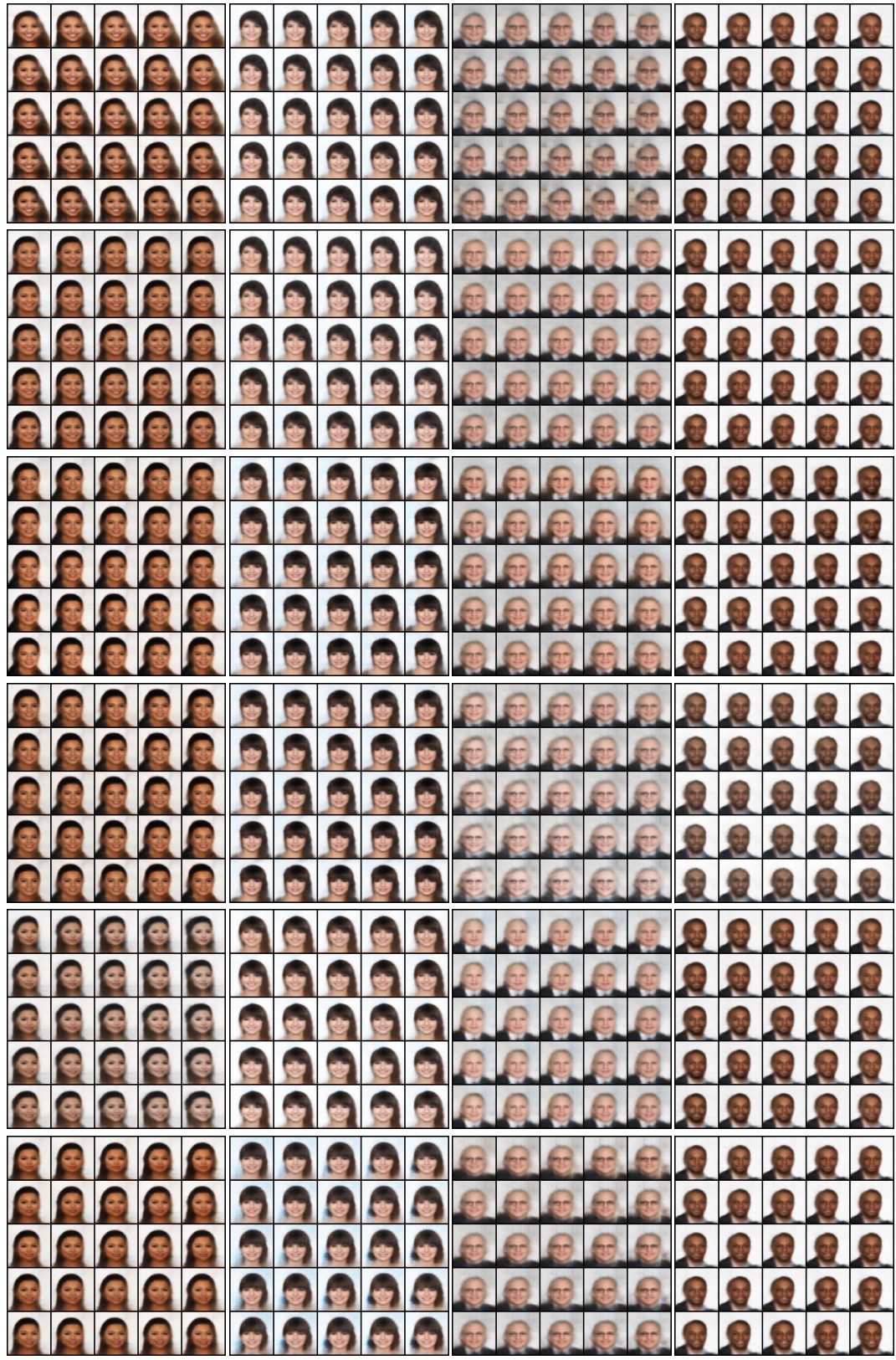

Figure 19: Various latent traversals for DIVA.

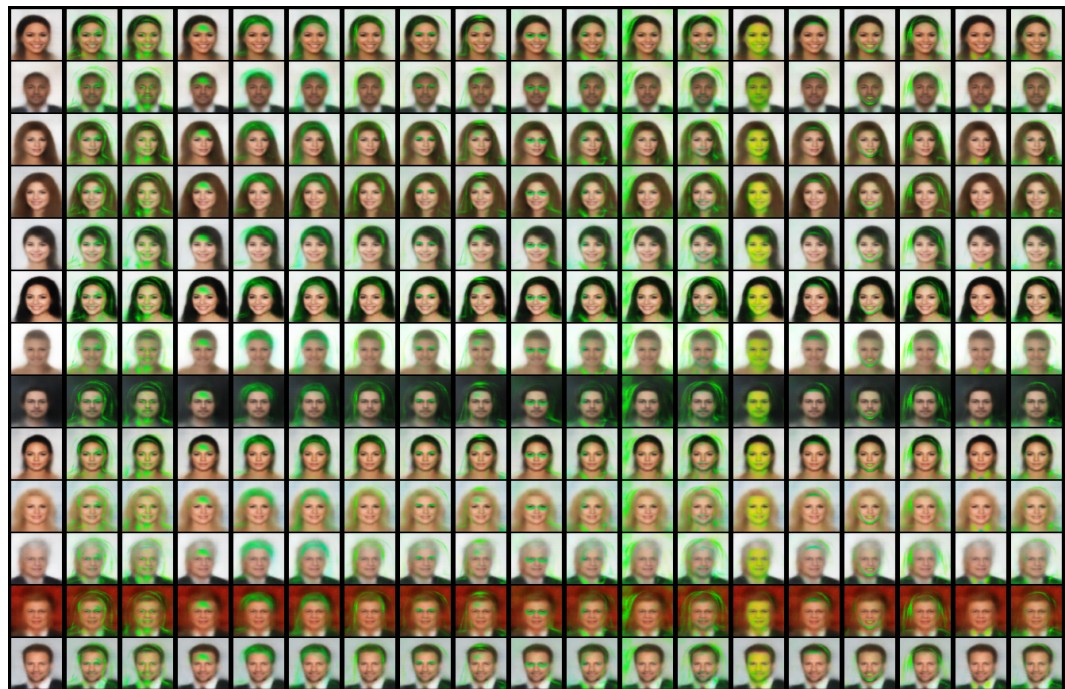

Figure 20: CCVAE, variance in reconstructions when intervening on a single label. From left to right: reconstruction, then interventions from switching on the following labels: `arched eyebrows`, `bags under eyes`, `bangs`, `black hair`, `blond hair`, `brown hair`, `bushy eyebrows`, `chubby`, `eyeglasses`, `heavy makeup`, `male`, `no beard`, `pale skin`, `receding hairline`, `smiling`, `wavy hair`, `wearing necktie`, `young`.

Table 6: Additional classification accuracies.

| Model | MNIST | | | | FashionMNIST | | | |
|---|---|---|---|---|---|---|---|---|
| | $f = 0.004$ | $f = 0.06$ | $f = 0.2$ | $f = 1.0$ | $f = 0.004$ | $f = 0.06$ | $f = 0.2$ | $f = 1.0$ |
| CCVAE | **0.927** | **0.974** | **0.979** | **0.988** | 0.741 | **0.865** | **0.879** | **0.901** |
| M2 | 0.918 | 0.962 | 0.968 | 0.981 | **0.756** | 0.848 | 0.860 | 0.892 |

**Conditional Generation** We provide classification accuracies for pre-trained classifier using conditional generated samples as input and the condition as a label. We also report the mutual information to give an indication of how *out-of-distribution* the samples are. In order to estimate the uncertainty, we transform a fixed pre-trained classifier into a Bayesian predictive classifier that integrates over the posterior distribution of parameters $\omega$ as $p(\boldsymbol{y} \mid \boldsymbol{x}, \mathcal{D}) = \int p(\boldsymbol{y} \mid \boldsymbol{x}, \omega) p(\omega \mid \mathcal{D}) \mathrm{d}\omega$. The utility of classifier uncertainties for out-of-distribution detection has previously been explored Smith & Gal (2018), where dropout is also used at test time to estimate the mutual information (MI) between the predicted label $\boldsymbol{y}$ and parameters $\omega$ (Gal, 2016; Smith & Gal, 2018) as

$$I(\boldsymbol{y}, \omega \mid \boldsymbol{x}, \mathcal{D}) = H[p(\boldsymbol{y} \mid \boldsymbol{x}, \mathcal{D})] - \mathbb{E}_{p(\omega \mid \mathcal{D})} \left[ H[p(\boldsymbol{y} \mid \boldsymbol{x}, \omega)] \right].$$

However, the Monte Carlo (MC) dropout approach has the disadvantage of requiring *ensembling* over multiple instances of the classifier for a robust estimate and repeated forward passes through the classifier to estimate MI. To mitigate this, we instead employ a sparse variational GP (with 200 inducing points) as a replacement for the last linear layer of the classifier, fitting just the GP to the data and labels while holding the rest of the classifier fixed. This, in our experience, provides a more robust and cheaper alternative to MC-dropout for estimating MI. Results are provided in Table 7.

**Latent Traversals** We can also perform latent traversals for the multi-class setting. Here, we perform linear interpolation on the polytope where the corners are obtained from the network $\boldsymbol{\mu}_\psi(\boldsymbol{y})$ for four different classes. We provide the reconstructions in Figure 21.

Table 7: Pre-trained classifier accuracies and MI for MNIST (top) and FashionMNIST (bottom).

| | Model | $f = 0.004$ | | $f = 0.06$ | | $f = 0.2$ | | $f = 1.0$ | |
|---|---|---|---|---|---|---|---|---|---|
| | | Acc | MI | Acc | MI | Acc | MI | Acc | MI |
| M | CCVAE | **0.910** | **0.020** | **0.954** | **0.014** | **0.961** | **0.013** | **0.973** | **0.010** |
| | M2 | 0.883 | 0.035 | 0.929 | 0.026 | 0.934 | 0.024 | 0.948 | 0.020 |
| F | CCVAE | 0.734 | **0.025** | **0.806** | **0.024** | **0.801** | 0.028 | **0.798** | **0.029** |
| | M2 | **0.750** | 0.032 | 0.792 | 0.032 | 0.787 | 0.032 | 0.789 | 0.031 |

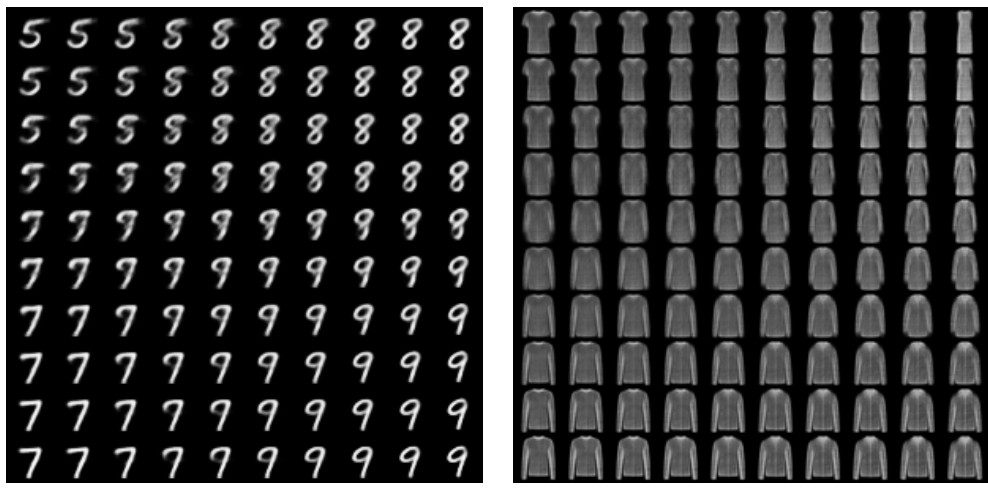

Figure 21: CCVAE latent traversals for MNIST and FashionMNIST. It is interesting to see how one class transforms into another, e.g. for MNIST we see the end of the 5 curling around to form an 8 and a steady elongation of the torso when traversing from `t-shirt` to `dress`.

**Diversity in Conditional Generations**    Here we show how we can introduce diversity in the conditional generations whilst keeping attributes such as pen-stroke and orientation constant. Inspecting the M2 results Figure 22 and Figure 23, where we have to sample from $z$ to introduce diversity, indicates that we are unable to introduce diversity without affecting other attributes.

**Interventions**    We can also perform interventions on individual classes, as showed in Figure 24.

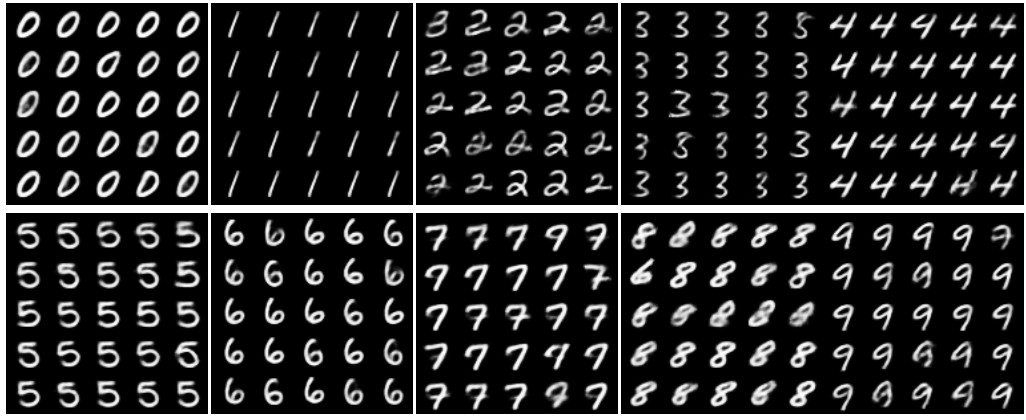

Figure 22: CCVAE conditional generations with $z_{\backslash c}$ fixed. Here we can see that CCVAE is able to introduce diversity whilst preserving the "style" of the digit, e.g. pen width and tilt.

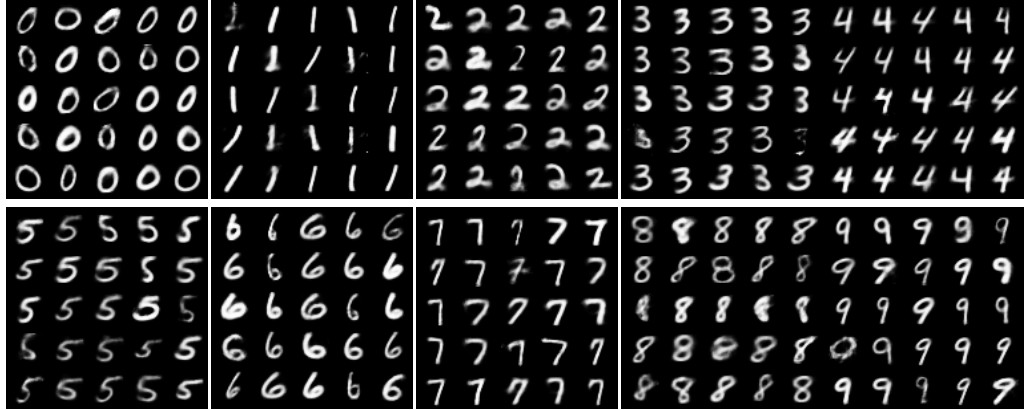

Figure 23: M2 conditional generations. Here we can see that M2 is unable to introduce diversity without altering the "style" of the digit, e.g. pen width and tilt.

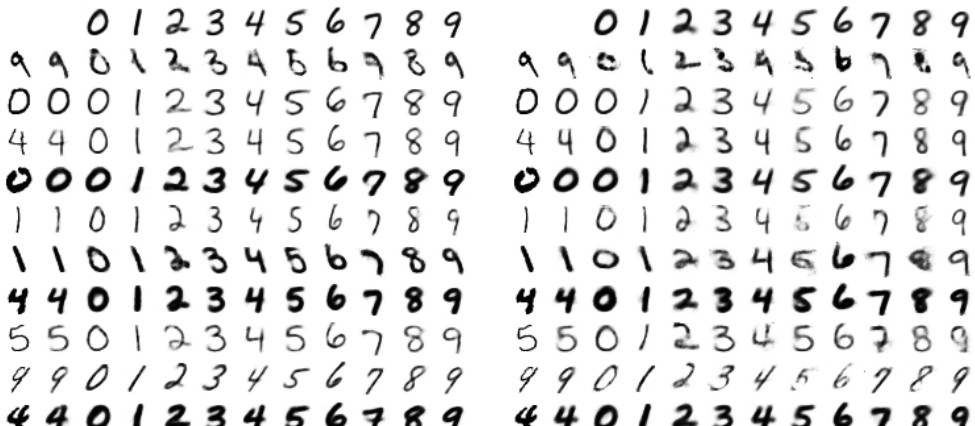

Figure 24: Left: CCVAE, right: M2. As with other approaches, we can also perform wholesale interventions on each class whilst preserving the style.

