# OpenReview forum: "Capturing Label Characteristics in VAEs"
_ICLR.cc/2021/Conference — ICLR 2021 Poster_

### Official Review · AnonReviewer3 · 2020-10-18
**Interesting paper, but there are some needed clarifications**

**Rating:** 6
**Confidence:** 4

**Review:**

Update after rebuttal: the authors have clarified some of my concerns, and I have therefore increased my score.

---------------
This paper introduces the characteristic-capturing VAE, an architecture that extends VAEs by modelling label-dependent characteristics in generative and classification tasks. In a multi-label setting, the CCVAE learns representations of individual characteristics, which are disentangled by design to the one of other labels.

The paper is well written in general, and is a good contribution to the important research direction on deep generative models focusing on using some labelled data to learn a disentangled latent space.
The experiments in the paper are well thought and show convincing performances in terms of label-dependent generations and interventions.

My biggest doubts are in terms of the positioning/novelty of the paper, which I found a bit confusing/misleading in the current form. When "rethinking supervision" the authors argue for the need of treating labels as auxiliary variables - and not as latent variables - in the model. This leads to the formulation called ALVAE, that introduces label-dependent information in latent space learning jointly a classifier from z to y. This formulation is novel as far as I know, but leads to some challenges mentioned by the authors in terms of conditional generation/intervention with this model. The authors then offer as a solution the CCVAE, which uses a conditional generative model p(z_c|y)p(y).
* Is y in a CCVAE still an auxiliary variable? To me it seems that the benefits of the CCVAE are not due to the auxiliary nature of y, but more from the fact that y is now a partially observed latent variable that is however only connected to x through continuous latent variables. If this is the case, why is it relevant to leave in the paper the whole ALVAE discussion which is purely theoretical and not even presented in the experiments? Perhaps the authors should focus more instead on clarifying even further on the benefits of the hierarchical generative architectures y->z->x.
* The discussions in section 3 are valid for 1-layer models such as M2 or MVAE models in which z_y=y. This discussion is not however considering hierarchical VAE architectures for semi-supervised learning, such as the one presented in "Semi-Supervised Generation with Cluster-aware Generative Models" by Maaløe et al, 2017. These architectures also learn a latent space that captures the information associated with a class, not just the class itself, and do so without the need for "auxiliary" variables. Is there any advantage of the CCVAE has with respect to this model?

---

> ### Author Response · Authors · 2020-11-19
> **Response to Reviewer 3 (2/2)**
>
> * ”_The discussions in section 3 are valid for 1-layer models such as M2 or MVAE models in which_ z_y=y. _This discussion is not however considering hierarchical VAE architectures for semi-supervised learning, such as the one presented in "Semi-Supervised Generation with Cluster-aware Generative Models" by Maaløe et al, 2017. These architectures also learn a latent space that captures the information associated with a class, not just the class itself, and do so without the need for "auxiliary" variables_".
> The connection to hierarchical semi-supervised VAEs is indeed an interesting one, and something we are keen to explore going forward in the context of our model. Thank you for your pointer to Maaløe et al., 2017 [1].
> It is important to note however that where such models employ semi-supervised learning [1,2], the intention appears to be to incorporate labels in much the same fashion as M2 by treating labels as a latent in addition to the hierarchy of $\textbf{z}_i$ variables, see Figure 2 in [1] and Appendix F in [2], for an example.
> While the latent variables do capture some notion of hierarchy information from the data (eg. Figure 5 of [1]), there is no constraint to disentangle the characteristic information of the label (what we refer to as $\textbf{z}_c$) from information in the data that is independent of all labels (what we refer to as $\textbf{z}_\c$).
> It would indeed be an interesting question to explore how using $\textbf{y}$ as an auxiliary variable applies in the hierarchical latent case, but that is beyond the scope of this work. We have included references to hierarchical models in our updated manuscript.
> [1] Semi-Supervised Generation with Cluster-aware Generative Models, Maaløe et al., 2017
> [2] BIVA: A Very Deep Hierarchy of Latent Variables for Generative Modeling, Maaløe et al., 2019
>
>
> * _“Is there any advantage of the CCVAE has with respect to this model?”_:
> Yes indeed:
> a. CCVAE is simpler; it doesn't require extra latent variables
> b. CCVAE has a sound objective that targets the actual variational free energy whereas CaGeM employs additional regularisers with
> associated hyperparameters.
> c. CCVAE can perform explicit manipulations of characteristics such as characteristic swaps (Figure 7); something that CaGeM cannot do because it does not have latent variables independent of labels
> d. CCVAE demonstrates its utility and performance on more complex data such as CelebA and Chexpert

---

> ### Author Response · Authors · 2020-11-19
> **Response to Reviewer 3 (1/2)**
>
> We would like to thank you for your praise and constructive suggestions for improving the paper. We would specifically like to acknowledge your positive words with relation to the contribution to the field and the quality of the experiments. We hope the following response addresses your concerns.
>
>
> In terms of the novelty/positioning of the paper. We first would like to point out there are two key aspects to our work: 1) Outlining why placing labels directly in the latent space is not a good idea and explaining how we can avoid this (a significant conceptual/theoretical innovation). 2) Introducing the novel CCVAE method which is able to realize this and perform various tasks previous approaches cannot. We have presented the paper in a way that tries to separately convey these two contributions and we feel that removing either would substantially weaken the paper.
>
> * _“Is y in a CCVAE still an auxiliary variable?”_:
> Yes. In CCVAE $\textbf{y}$ remains an auxiliary variable which we partially observe. We hoped to illustrate why this is a sensible approach through the introduction of ALVAE (now renamed to Eq 2), we have updated the paper to re-emphasize that in CCVAE $\textbf{y}$ remains an auxiliary variable.
>
> * _“To me it seems that the benefits of the CCVAE are not due to the auxiliary nature of y, but more from the fact that y is now a partially observed latent variable”_:
> We believe this concern may stem from a misunderstanding of what we mean when we denote $\textbf{y}$ as an auxiliary variable: we mean that it is only connected through another variable and is not directly used in reconstruction.  Critically, this means it can also be side-stepped during intervention, allowing us to perform more fine-grained interventions than just label swaps by adjusting the relevant ${z}_c^i$ (e.g. Figures 3, 4, and 7, Table 2, many of experiments in appendices); we could not do these adjustments if each $\textbf{y}^i$ was not an auxiliary variable.
>
> * _“why is it relevant to leave in the paper the whole ALVAE discussion which is purely theoretical and not even presented in the experiments?”_:
> The contribution of the paper goes far beyond a single algorithmic approach: it provides substantial new insights into how we should think about supervision in VAEs. ALVAE is purely introduced to demonstrate this new line of thought and how one might wish to go about structuring the latent space, forming a stepping stone from which CCVAE builds. We have subsequently updated the beginning of Section 4 and removed the name ALVAE to further delineate that it is a conceptual/theoretical contribution and key intermediary step towards CCVAE, rather than an algorithm we are suggesting should be used directly.  We emphasize though that we feel that it is a real strength of the paper to have both the important conceptual contributions that ALVAE conveys, while further having a more algorithmically orientated extension of these to allow their effective use in practice (via CCVAE).
>
> * _“Perhaps the authors should focus more instead on clarifying even further on the benefits of the hierarchical generative architectures y->z->x”_:
> Our model is not a hierarchical VAE in the conventional manner.  For example, when doing reconstructions we do not go through $\textbf{y}$.  As explained earlier, the key advantage of this is that it allows $\mathbf{z}$ to capture richer information than $\textbf{y}$ and that this, in turn, allows us to make more fine grained and nuanced interventions than previous setups.  We have updated Section 4.1 to highlight this.

---

> > ### Author Response · Authors · 2020-11-24
> > **Clarrification and paper update**
> >
> > To notify you directly, we've updated the paper to address your concerns. We have removed references to treating as an auxiliary variable and hope that our updates alleviate the ambiguity that the term introduced.
> >
> > Thank you for your help in improving the submission.
> >
> > We are also reposting this, as there might have been an issue with you receiving our initial comment:
> >
> > > Thanks for following up! Your description here is correct and we think it is simply a question of semantics on the word auxiliary: we believe our use is still consistent with more classical statistics and are using it to convey its relationship to z_c from the perspective of representation learning, but agree that it is maybe not a great match to how the term is often used in DGMs. We are happy to change this to make sure there is no ambiguity and will update the paper shortly. Did our responses clear up your other concerns?

---

### Official Review · AnonReviewer4 · 2020-10-25
**Capturing Label Characteristics in VAEs**

**Rating:** 5
**Confidence:** 4

**Review:**

In this paper the authors have propsed a method to incorporate label information
in variational autoencoders (VAEs) to captures the characteristics associated with
those labels. Experiments show that using the label information helps them to better
intervation and conditional generation of images.

The paper is well written. However I have few concern regarding the model.

Firstly the paper has not cited a very important related work by
Adel, Tameem, Zoubin Ghahramani, and Adrian Weller. "Discovering interpretable
representations for both deep generative and discriminative models." International
Conference on Machine Learning. 2018.
where a similar hiererchical approach was proposed.
The authors should provide a discussion related to this method or if possible should
compare their performance by extending the previous model for multiple features.

The main concerns regarding the model perspective are as follows:

1. How the authors ensuring that there is no mutual information between the parts z_c and z_\c ?
There is no analysis regarding the same. They should report some quanitative measures as proposed by
betaVae to support the same.

2. How they are controlling the KLD between encoding and decoing distribution i.e
KL(q(z|x) || p(z| z_c)) and KL(q(z_c|z) || p(z_c|y)) ? If the first kl is not significantly low,
there will be information loss which might affect the quality of the overall generation?
They should report the KL achieved in each layer.

3. As they are learning the reverse mapping from a given label to corresponding
representation i.e. p_\theta(z_c | y) --
is it necessary to make z_c unidimensional for interpolation ?
How are they learning the correspondance between fin grained values of y
and z_c ? do they have such label information available?

---

> ### Author Response · Authors · 2020-11-19
> **Response to Reviewer 4 (2/2)**
>
> _“Adel et al 2018”_
> Thank you for bringing the work of Adel et al. to our notice; the JLVM model in particular is indeed relevant. We however note that while their graphical model bears some close resemblance to ours, there are quite distinct differences between their and our approaches.
>
>  Model: The JLVM model can be viewed as a VAE with a normalising flow-based posterior that additionally includes side-channel information to influence the latent $z^*$. From that perspective, the JLVM model is closer to the MVAE model (Wu & Goodman, 2018) than to ours, as the conditional distribution $p(z^* | s)$ is viewed as a posterior rather than a prior (and vice versa with $p(s | z^*)$).
>
> Moreover, given the difference in information content between $x$ and $s$, having both project to exactly the same latent space $z^*$ results in an imbalance---something we explicitly account for by only jointly projecting to $z_c$, and leaving $z_\c$ to capture the residual information from $x$ \ $s$ (or y as we denote it), as shown in Figure 2.
>
> It is this explicit factorisation that allows us to perform 'characteristics swaps' (Figure  7), which would otherwise be infeasible as with the comparison models, and with the JLVM (or indeed the ILVM) model.\rObjective: For the JLVM model, the objective considered is also quite different in that it involves the (unweighted) concatenation of the total marginal likelihood over data and the (bounded) information bottleneck (IB) estimator. Note that for continuous latents $z^*$, this means that the objective is not guaranteed to target the variational free energy anymore---something our objective preserves.
>
> Moreover, the proposed objective has two potential hyperparameters to tweak: \beta corresponding to the IB margin, and the relative weight between the total marginal likelihood and the bounded IB estimator which is implicitly taken to be 1.0. Our objective on the other hand needs no extra hyperparameters.

---

> ### Author Response · Authors · 2020-11-19
> **Response to Reviewer 4 (1/2)**
>
> We would like to thank you for your hard work and constructive feedback. We hope that the following response helps alleviate your concerns.
>
> We start by responding to your 2nd listed concern as we believe that there has been a serious misunderstanding here that percolates to the other questions raised.
>
> * "[_How they are controlling the KLD KL_(q(z|x) || p(z| z_c)) _and KL_(q(z_c|z) || p(z_c|y))_? there may be information loss which might affect the overall generation_]":
> We believe this concern stems from a misunderstanding about our model.  There is only one level of latent variables and no hierarchy among them: ${z}_c$ is simply a subset of z (${z}=z_c\cup z_{\c}$) and so $p(z|z_c)$ nor $q(z_c|z)$ actually exist in the manner implied (see Figure 2) and cannot report the KLs you suggest.  ${z}_c$, and ${z}_\c$ vary only in how they relate to $y$: $q(z|x)$  and $p(x|z)$ are simple mappings, neither give any special treatment to ${z}_c$. As such, we never go through $y$ during reconstruction so there is no risk of information loss.
> Separately to the process of reconstruction, we are simultaneously learning a series of classifiers $q(y^i  | {z}_c^i)$ and pre-image distributions $p(z_c^i |y^i )$.  Here there is an inevitable, and deliberate, information loss because $y^i$  are discrete labels and ${z}_c^i$  continuous.  The core premise is that because our reconstruction relies on $z_c^i$ , it can capture this characteristic information that would otherwise have been lost (e.g. if using M2).  Critically, when performing interventions we can work with $z_c^i$  rather than with $y^i$  to ensure this information is preserved and allow richer interventions beyond simple label switches.
>
> * _“How [are] the authors ensuring that there is no mutual information between the parts z_c and z_\c ?”_:
> In general, the mutual information between $z_c$ and $z_\c$ (and between individual dimensions of $z_c$) will not be zero, nor should it be.  $z_c$ captures information *directly* related to a particular variable, but this does not mean that other information might not be indirectly related through correlations etc.  For example, doing an intervention that adds a beard to a person (cf Figure 18, penultimate row, second from left) should not also change their gender, but the mutual information between the two is clearly not zero.  Having a setup where the MI between $z_c$ and $z_\c$ is driven to zero would thus likely fail to actually achieve our goals due to inherent correlations in the data.
> Instead, our focus is to maximize the information that each $z_c^i$ contains about characteristics relating to $y^i$ through our latent space structuring and model setup.  This then ensures that manipulating $z_c^i$ provides the maximal desired effects.  Our empirical results confirm that this is successful and demonstrate that our interventions are not only effective, but do not induce unwanted effects on other variables (see e.g. Table 2, Figure 7, and the confusion matrices in the appendices).  This latter property is critical in the context of your concern as it is exactly what one might be worried about from information sharing between $z_c$ and $z_\c$.
>
> *  "_is it necessary to make_ z_c _unidimensional for interpolation ?_":
> Though it is not necessary from the perspective of classification or conditional generation, having each $z_c^i$ be unidimensional is helpful for performing interpolations.  Namely, though in principle they can be multidimensional, it is then difficult to know how to intervene in a way that induces the desired effects.  We have generally found that allowing each $z_c^i$ to only relate to a single class label is perfectly sufficient without meaningfully restricting model fidelity.
>
> * "_How are they learning the correspondence between fin grained values of $y$ and_ $z_c$ _? do they have such label information available?_": The only supervised information we have available is a vector of labels, there is no more fine-grained information from the labels themselves.  Each label y^i corresponds in a one-to-one mapping with an individual $z_c^i$ (as per Figure 2).  This structuring is then what enforces the correspondence between $z_c^i$ and $y^i$, as the only information available in the classification of $y^i$ is $z_c^i$.  Any more fine grained information we pick up is directly as a result of this structuring allowing us to exploit additional information in the input data.

---

### Official Review · AnonReviewer2 · 2020-10-27
**A new VAE-based framework for utilizing context information**

**Rating:** 7
**Confidence:** 5

**Review:**

**GENERAL**
The paper proposes to re-think the fashion of using label information in the VAE framework. The authors propose to disentangle information about the label (or, more generally, the context) in a "hard-coded" manner, namely, by using a separate set of variables for the label (context). The paper is written in a lucid manner, and the presented results are sound.

**Strengths:**
S1: The presented VAE framework (CCVAE) is interesting.

S2: The final objective in Eq. (6) that follows from applying Jensen's inequality two times, allows to efficiently train the model. Moreover, it allows the use of interventions to modify information about images.

S3: The demo is a nice fashion of presenting the idea.

S4: I highly appreciate that the authors used a dataset outside of standard benchmarks, and they focused on a medical application (the Chexpert dataset).

S5: The authors explained all implementation details that increases reproducibility of the paper.

S6: I appreciate the close relation of the propose framework to other models, e.g., DIVA. Moreover, the idea of improving upon DIVA is very interesting.

**Deficiencies:**
D1: My only concern is rather average generative capabilities of the model. Nevertheless, it does not affect my overall good assessment of the paper.

**Remarks:**
R1: In Eq. 2, there is a fudge factor, \alpha. However, in the CCVAE objective (Eq. 4) it is no longer necessary. I would like to ask the authors whether they thought about adding this fudge factor, or it is simply unnecessary. (I am aware that q(y|x) follows naturally from the objective, nevertheless, it might further improve the classification accuracy).

---

> ### Author Response · Authors · 2020-11-19
> **Response to Reviewer 2**
>
> We would like to thank you for your hard work and positive feedback.  In particular, we appreciate the effort you have undertaken to understand and convey the nuances and underlying concepts of our paper that set it apart from other work.
>
>
> * _“[Fudge factor]”_:
> The idea of a fudge factor is an interesting one and something we did actually consider, but generally found provided little improvement.  Our setup effectively allows for a natural scaling of the classification pressure and the fact that we achieve such good results without such heuristics or hyperparameters is, in our opinion, a real strength of the paper. We did find that improving the flexibility of the classifier helped improve the accuracy for the MNIST/FashionMNIST experiments, see Appendix E.5. We believe that this is because in contrast to other approaches our objective corresponds to the variational free energy of the model.

---

> > ### Comment · AnonReviewer2 · 2020-11-23
> > **After the rebuttal**
> >
> > Dear authors,
> >
> > Thank you for your comment. Honestly, I expected that the fudge factor should have a little influence, but it's great to hear your confirmation.
> >
> > In my opinion, the paper deserves to be accepted and, thus, I keep my score (7).
> >
> > Best.

---

### Official Review · AnonReviewer1 · 2020-10-28
**The relationships to some existing methods are not sufficiently discussed.**

**Rating:** 6
**Confidence:** 4

**Review:**

This paper focuses on latent representations learning when some labels are provided. The authors propose a method called the characteristic capturing VAE (CCVAE). This method learns real-valued auxiliary variables that capture the label information. The proposed method is tested on a medical image and a face image dataset.

One of the major motivations of the proposed method is that it can be used to conditionally generate images based on desired characteristics. However, this paper proposes a VAE model, which usually generates lower-quality images compared to Generative Adversarial Networks (GAN). It is not clear to me why this paper focuses on a VAE model rather than a GAN model. There have been GAN-based methods [1, 2] also based on an auto-encoding framework. Note that if we remove the adversarial loss from the objective functions of these methods, and add the KL divergence penalty, these methods become VAE. Since these methods also introduce real-value variables, I believe the authors should compare to the VAE-version of these methods.

In addition, in the VAE literature, [3] proposes a semi-supervised method.  Since [3] also involves a real-value latent variable, it looks like the difference between the proposed method and [3] is that the author extends [3] to multiple labels. Is this true? I suggest the authors better clarify the novelty of the proposed method compared to [3].

I do not suggest accepting this paper because the relationships to some existing methods are not sufficiently discussed.

Minor:
In the experiments, the paper reports the quantitative measures for classification and disentanglement. However, no quantitative measures for image quality are reported. I suggest the authors report some measures such as reconstruction error and FID score.

References
[1] Xiao, Taihong, Jiapeng Hong, and Jinwen Ma. "DNA-GAN: Learning disentangled representations from multi-attribute images." International Conference on Learning Representation Workshop. 2018.

[2]Xiao, Taihong, Jiapeng Hong, and Jinwen Ma. "Elegant: Exchanging latent encodings with gan for transferring multiple face attributes." Proceedings of the European conference on computer vision (ECCV). 2018.

[3] Li, Yang, et al. "Disentangled variational auto-encoder for semi-supervised learning." Information Sciences 482 (2019): 73-85.

---

> ### Author Response · Authors · 2020-11-19
> **Response to Reviewer 1**
>
> We would like to thank you for your review and effort to help us improve our submission. We hope our response below (and associated paper update) alleviates some of your concerns and hope you will consider increasing your score if it does.  In particular, we would ask you to consider the fact that large numbers of papers on VAEs are published at ICLR each year, such that the fact the paper is based around VAEs should not itself be a reason for rejection.
>
> * _“It is not clear to me why this paper focuses on a VAE model rather than a GAN model”_:
> The primary contribution of this submission is demonstrating how labels should be used to obtain interpretable/manipulatable representations, for which VAEs are a suitable and popular model.  Though GANs tend to have better generative capabilities, they are not generally considered superior from a representation learning perspective (and arguably are actually less effective here).  Moreover, many aspects of both our conceptual contributions and our algorithmic approach would not be directly applicable to a GAN setting; our contributions to the literature would not be possible without taking a VAE approach.  Indeed, the flexibility of the VAE framework to allow innovations like ours are one of its greatest strengths.
>
> * _“I do not suggest accepting this paper because the relationships to some existing methods are not sufficiently discussed.”_:
> Though we have gladly added references and a short discussion to the paper on these works, we point out that they are of less relevance to the work than many more closely related papers already discussed.  For example, [1] and [2] are based on a completely different model class and do not allow the same range of tasks to be performed.  [3] is an extension to M2, which is already discussed extensively, and its extensions relative to M2 do not themselves seem to relate to our contributions.
> Taking this into account, we do not feel that this is a justified reason for rejecting the paper, particularly now the citations and discussion have been added, and ask the reviewer to reconsider their score accordingly.
>
> * _“I believe the authors should compare to the VAE-version of [[1] and [2]]”_:
> This is unfortunately not feasible.  Such methods do not currently exist in the literature and would be novel contributions in their own right.  Moreover, they are less relevant baselines than those already compared to.
>
> * _“I suggest the authors report some measures such as reconstruction error and FID score.”_:
> We would like to point out that there are the more relevant generative classification accuracies presented in the Appendix along with MI scores for MNIST and FashionMNIST: unconditional generation is not the aim of this work and FID is far from an absolute measure of visual fidelity, meaning it is of little relevance to our aims.  Nonetheless, we have added FID scores as per your request to Appendix E.3, where we see that these are almost identical for CCVAE and M2 as one would expect. We opted not to report reconstruction error as this metric provides little intuition regarding the disentangled nature of the latent space and adds little beyond the FID scores themselves. We can still add this if you think it is necessary though.

---

> > ### Comment · AnonReviewer1 · 2020-11-23
> > **Thank you for your response.**
> >
> > If I understand correctly, this paper focuses on conditional image generation, making use of label information. There are such methods based on GAN. It looks like GAN generates higher-quality images than the proposed VAE method. Therefore, I believe the paper would be much stronger if the authors can provide situations where the proposed VAE method outperforms GAN. Note that a large amount of VAE papers are published because it helps the downstream tasks because it introduces disentanglement or fairness, etc. during training, rather than focusing on image generation. Unfortunately, I do not observe any evidence showing that the proposed VAE method helps in any downstream tasks.
> >
> > In addition, the GAN-based methods in [1] and [2] have introduced some ideas related to the auto-encoding framework, including making use of label information by swapping latent encodings and improving imaging reconstruction by learning the residual images. I do not observe any evidence that the proposed VAE strategy is superior to these ideas, and it is not clear to me why the proposed method is novel compared to these ideas. It also looks relatively straightforward to implement the VAE version of these methods.  One can remove the adversarial loss from the objective functions, and add the KL divergence penalty. Without comparison, I am not convinced that the proposed method is novel.

---

> > > ### Author Response · Authors · 2020-11-23
> > > **Response 2 to Reviewer 1**
> > >
> > > Thank you for following up.  However, we believe you have quite significantly misunderstood the paper: our aim is not simply conditional image generation which we directly point out in the previous response and also outline many times in the paper along with various experiments on different tasks. In particular, we do consider downstream tasks, such as classification, and this is exactly why a GAN is not appropriate as per the arguments you layout yourself.  We simply cannot compare to the methods you suggest because they are only able to accomplish a small number of the tasks we consider.  Moreover, we cannot understand why the experimental comparisons you suggest relates to the novelty of our work, even if we could make such comparisons.  We note that none of the more closely related baselines we do compare to make comparisons to methods like those you allude to, or indeed even cite or discuss them.

---

> > > > ### Comment · AnonReviewer1 · 2020-11-23
> > > > **Could you simply give me the list of tasks?**
> > > >
> > > > Could you simply give me the list of tasks which can be achieved by the proposed method, but cannot be done by the GAN as described in [1] and [2], and explain why? That is what I am asking. Note that the GAN-based methods cannot generate good results if they do not capture the labeling characteristics. In other words, those methods also capture label characteristics. Adversarial loss simply improves the image qualities. The methods still work properly if you remove the adversarial loss from the objective function, and they become traditional autoencoders.
> > > >
> > > > There exists another strategy to capture label characteristics in the auto-encoding framework. Why is it irrelevant to your purposes?

---

> > > > > ### Author Response · Authors · 2020-11-24
> > > > > **Differences between CCVAE and [1, 2]**
> > > > >
> > > > > There are fundamental differences in the approach and the aims taken in [1,2] compared to our proposed method. For one, [2] only considers something resembling what we call "interventions" (referred to as generation by exemplars in [2]). This is understood as swapping characteristics of one picture with another. Secondly, they only consider the _fully_ supervised case and not arbitrary supervision rates like we do. And lastly, our approach achieves all of: characteristic swaps, single interventions, multiple interventions, conditional generations, classification, while also deriving a novel objective that faithfully represents the variational free energy of the model.
> > > > > To be explicitly clear, the features our model can perform that [1, 2] cannot is:
> > > > > * Semi-supervised. There is no mention in [1, 2] on what to do if the labels are missing, whereas in our work we explicitly outline how to deal with this case and indeed forms a significant part of our novel contribution.
> > > > > * Intervention on a single image. We are able to form interventions on a single image and do not require pairs like in [1, 2].
> > > > > * Conditional generation, i.e. generating images without the need for exemplars. Due to the fact we obtain a stochastic representation, we are able to generate images without the need for exemplars like in [1, 2].
> > > > > * Classification: a significant strength of CCVAE is the emergence of a natural classifier from our universal approach, which does not feature in [1, 2].
> > > > > * Our objective represents the variational free energy of the model, an attractive characteristic which the objectives of [1, 2] do not.
> > > > >
> > > > > In short, the closest experiment for which [1, 2] could be used as a baseline is the characteristic swaps (Fig 7), but only in the _fully_ supervised case, and as they do not allow for the unsupervised case they are not applicable. All of our other experiments: latent traversals, interventions on single images, conditional generations, classification, cannot be achieved by [1, 2], which as such, makes them an unsuitable baseline.

---

### Author Response · Authors · 2020-11-19
**General Response to Reviewers**

We would like to thank the reviewers for their hard work and constructive feedback. We would particularly like to thank them for the positive comments on: novelty of the submission (R2, R3); the comprehensive experiments (R2, R3, R4); introduction of demo application (in supplement) (R2); contribution to the field of generative models (R2, R3); and clarity (R2, R3, R4).

We have taken on board the concerns and suggestions raised by the reviewers and have made various updates to the paper to address them.  For example, we have added suggested citations and provided a discussion where relevant; added clarifications on how our model relates to hierarchical VAEs; added FID score results; and made general clarifying edits throughout the paper. We further believe some of the key concerns raised actually stem from misunderstandings about the approach, so ask that the reviewers also carefully consider our individual responses which we hope clarify these.

Thank you all for your time and consideration!

---

### Decision · Program_Chairs · 2021-01-07
**Final Decision**

**Decision:**

Accept (Poster)

**Comment:**

This paper proposes a novel technique to learn a disentangled latent space using VAEs and semi-supervision. The technique is based on a careful specification of the joint distribution where the labels inform a factorisation of the distribution over continuous latent factors. The technique allows for inference, generation, and intervention in a tractable way.

The paper is well-written, the formulation is original, and the experiments convincing. There were some confusions that were mostly resolved during the discussion.

In addition to the expert reviews attached, I would like to remark that I too find the formulation interesting and elegant. And if I may add to the discussion, oiVAE (output-interpretable VAEs) by Ainsworth et al presented at ICML18 is a related piece of work that did not occur to me earlier, but which the authors could still relate to (I'd certainly enjoy reading about the authors' views on that line of work).